# Early-career setback and future career impact

Yang Wang [1,2,3,4], Benjamin F. Jones[1,2,3,5] & Dashun Wang [1,2,3,6]

Setbacks are an integral part of a scientific career, yet little is known about their long-term effects. Here we examine junior scientists applying for National Institutes of Health R01 grants. By focusing on proposals fell just below and just above the funding threshold, we compare near-miss with narrow-win applicants, and find that an early-career setback has powerful, opposing effects. On the one hand, it significantly increases attrition, predicting more than a 10% chance of disappearing permanently from the NIH system. Yet, despite an early setback, individuals with near misses systematically outperform those with narrow wins in the longer run. Moreover, this performance advantage seems to go beyond a screening mechanism, suggesting early-career setback appears to cause a performance improvement among those who persevere. Overall, these findings are consistent with the concept that "what doesn't kill me makes me stronger," which may have broad implications for identifying, training and nurturing junior scientists.

[1] Center for Science of Science and Innovation, Northwestern University, Evanston, IL 60208, USA. [2] Kellogg School of Management, Northwestern University, Evanston, IL 60208, USA. [3] Northwestern Institute on Complex Systems, Northwestern University, Evanston, IL 60208, USA. [4] School of Public Policy and Administration, Xi'an Jiaotong University, Xi'an 710049, China. [5] National Bureau of Economic Research (NBER), Cambridge, MA 02138, USA. [6] McCormick School of Engineering, Northwestern University, Evanston, IL 60208, USA. Correspondence and requests for materials should be addressed to D.W. (email: dashun.wang@northwestern.edu)

"**S**cience is 99 percent failure, and that's an optimistic view", said Robert Lefkowitz, who was awarded the Nobel prize in 2012 for his groundbreaking studies of G protein-coupled receptors. Despite the ubiquitous nature of failures, it remains unclear if a setback in an early career may augment or hamper an individual's future career impact. Indeed, the Matthew effect[1–9] suggests a rich get richer phenomenon where early-career success helps bring future victories. In addition to community recognition, bringing future attention and resources[5,7,8,10–15], success may also influence individual motivation[16], where positive feedback bolsters self-confidence. Together, these views indicate that it is early-career success, not failure, that would lead to future success. Yet at the same time other mechanisms suggest that the opposite may also be true. Indeed, screening mechanisms[17,18] suggest that, if early-career failures screen out less-determined researchers, early setbacks among those who remain could, perhaps counterintuitively, become a marker for future achievement. Further, failure may teach valuable lessons that are hard to learn otherwise[19–21], while also motivating individuals to redouble effort[22,23], whereas success may be associated with complacency[16] or reduced future effort due to utility maximization[24]. Such positive views of failure are reflected in Nietzsche's classic phrase "what doesn't kill me makes me stronger"[25], in the celebration-of-failure mindset in Silicon Valley[26], and in a recent commencement address by U.S. Supreme Court chief justice John Roberts, who told graduating

students "I wish you bad luck." Overall, these divergent perspectives indicate that the net effect of an early-career setback is unclear. Given the consequential nature of this question to individual careers and the institutions that support and nurture them, and building on the remarkable progress in our quantitative understanding of science[1,2,7–9,27–51], here we ask: Can an early-career setback lead to future career impact?

To offer quantitative answers to this question, we leverage a unique dataset, containing all R01 grant applications ever submitted to the NIH, to examine early-career success and failure. NIH funding decisions are largely determined by paylines derived from evaluation scores. Our empirical strategy harnesses the highly nonlinear relationship between funding success and evaluation score around the funding threshold (Fig. 1a). Indeed, focusing on individuals whose proposals fell just above and below the threshold allows us to compare observationally-similar individuals who are either near misses (individuals who just missed receiving funding) or narrow wins (individuals who just succeeded in getting funded). Here we focus on junior scientists by examining principal investigators (PIs) whose first application to the NIH was within the previous three years. We combine the NIH grant database with the Web of Science data, tracing their NIH R01 grant applications between 1990 and 2005 together with research outputs by the PIs, measured by their publication and citation records (see Supplementary Note 1 for details). In total, our analyses yielded 561 narrow wins and 623 near misses around the payline.

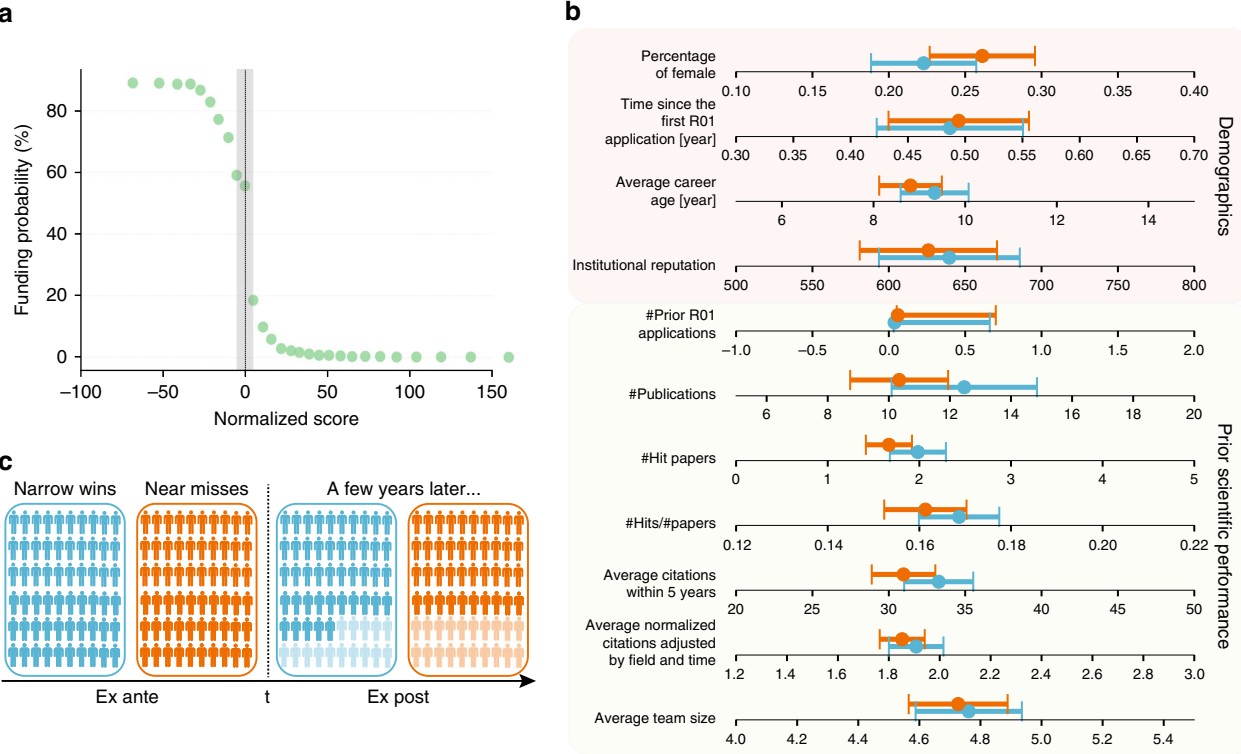

**Fig. 1** Pre-treatment comparisons between the narrow-win and near-miss applicants. **a** Relationship between normalized score and award status. Funding probability shows a clear transition around the funding threshold. We focus only on junior PIs whose normalized scores lie within the range (−5, 5), the shaded gray area, which includes 561 narrow-win and 623 near-miss applicants in our sample. **b** Pre-treatment feature comparisons between the near-miss and narrow-win group. We compared 11 different demographic and performance characteristics. The features are defined as follows (from top to bottom): (1) percentage of female applicants; (2) number of years since the first R01 application; (3) number of years since the first publication; (4) institutional reputation, measured by the number of R01 grants awarded to an institution between 1990 to 2005; (5) number of previous R01 applications; (6) number of publications prior to treatment; (7) number of prior papers that landed within the top 5% of citations within the same field and year; (8) probability of publishing a hit paper; (9) average citations papers received within 5 years of publication; (10) citations normalized by field and time;[34] and (11) average team size across prior papers. We see no significant difference between the two groups across any dimension we measured; Error bar represents the 95% confidence interval. **c** An illustrative example of the underlying process. Solid color indicates people who remained active, whereas shaded color denotes the fraction that disappeared from the NIH system. Blue and orange indicate narrow-win and near-miss applicants, respectively

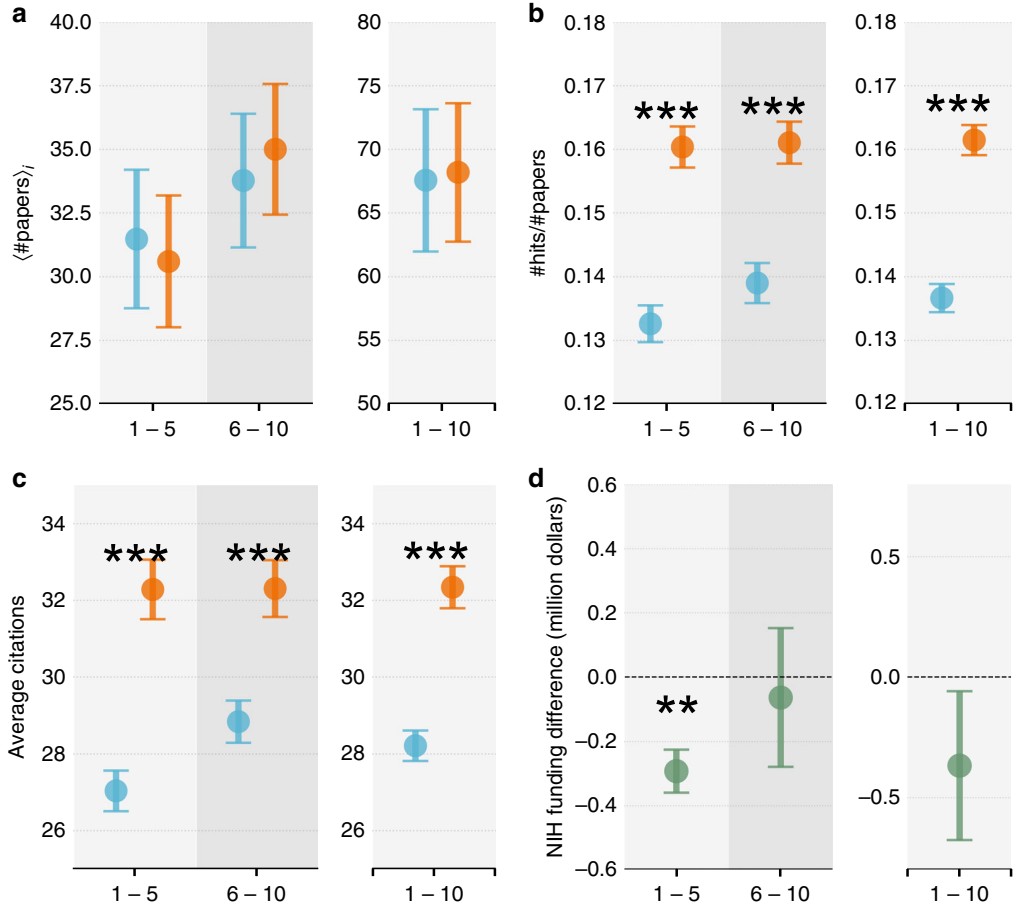

**Fig. 2** Comparing future career outcome between near misses (orange) and narrow wins (blue). **a** The average number of publications per person. **b** Near misses outperformed narrow wins in terms of the probability of producing hit papers in the next 1–5 years, 6–10 years, and 1–10 years. Note that there appears a slight performance improvement for the narrow-win group in the second five-year period, but the difference is not statistically significant ($\chi^2$-test $p$-value > 0.1, odds ratio = 1.05). **c** Average citations within 5 years of publication. The near-miss applicants again outperformed their narrow-win counterparts. To ensure all papers have at least 5 years to collect citations, here we used data from 1990 to 2000 to avoid any boundary effect. **d** Funding difference between the near-miss and narrow-win group from the NIH (near misses minus narrow wins). ***$p$ < 0.001, **$p$ < 0.05, *$p$ < 0.1; Error bars represent the standard error of the mean

We examine performance and demographic characteristics for the two groups of PIs, finding that prior to treatment, they are statistically indistinguishable along all dimensions we measured (Fig. 1b). Yet the treatment created a clear difference between the two, whereby one group was awarded R01 grants, which on average amount to $1.3 million for five years, while the other group was not. Given the pre-treatment similarity between the two groups, we ask: which group produced works with higher impacts over subsequent years?

## Results

**Future career impact.** We therefore traced the publication records of PIs from the two groups. We first focus on active PIs in the NIH system, defined as those who apply for and/or receive NIH grants at some point in the future (Fig. 1c and 'Different definitions of active PIs' in Supplementary Note 3). We calculated the publication rates of the PIs, finding that the two groups published a similar number of papers per person over the next ten-year period (Fig. 2a), consistent with prior studies[8,15,52]. We then computed, out of the papers published by the near-miss and narrow-win group, the probability of finding hit papers (Fig. 2b), defined as being in the top 5% of citations received in the same year and field (as indicated by the Web of Science subject category)[34,37]. In the first five years, 13.3% of papers published by the

narrow-win group turned out to be a hit paper, which is substantially higher than the baseline hit rate of 5%, demonstrating that narrow wins considered in our sample produced hit papers at a much higher rate than average scientists in their field. We measured the same probability for the near-miss group, finding that they produced hit papers at an average rate of 16.1%. Comparing the two groups, we find near misses outperformed narrow wins significantly, by a factor of 21% ($\chi^2$-test $p$-value < 0.001, odds ratio = 1.25). This performance advantage persisted: We analyzed papers produced in the second five-year window (6–10 years after treatment), uncovering a similar gap (Fig. 2b, $\chi^2$-test $p$-value < 0.001, odds ratio = 1.19). To ensure the observed effect is not just limited to hit papers, we also quantified performance using other commonly used measures, including average citations received within five years of publication (Fig. 2c) and the relative citation ratio (RCR) of each paper (see 'Normalized citations over time and disciplines' in Supplementary Note 3)[53,54], arriving at the same conclusions. Indeed, papers published by the near-miss group in the next two five-year periods attracted on average 19.4% (32.3 for near misses and 27.0 for narrow wins, $t$-test $p$-value < 0.001, Cohen's $d$ = 0.08) and 12.0% more citations (32.3 for near misses and 28.8 for narrow wins, $t$-test $p$-value < 0.001, Cohen's $d$ = 0.06) than those by the narrow-win group, respectively (Fig. 2c).

To further test the robustness of our results we repeated our analyses along several dimensions. We changed our definitions of junior PIs to two alternatives, by focusing on first-time R01 applicants only and by restricting to those without any current NIH grants ('Alternative definitions of junior PIs' in Supplementary Note 3). We varied our definitions of hit papers (from top 1% to top 15% of citations, 'Varying thresholds for the definitions of hit papers' in Supplementary Note 3). We computed per capita measures of hit papers ('Hits per capita' in Supplementary Note 3). We adjusted for field differences of citations, by calculating the average normalized citations by field and year[34] ('Normalized citations over time and disciplines' in Supplementary Note 3). We also varied our definition of fields using the Medical Subject Headings (MeSH)[55] ('Different field definition' in Supplementary Note 3). We repeated our analyses across different measurement time periods ('Robustness to alternative fiscal years' in Supplementary Note 3). We also checked whether the results may be affected by pre-existing papers moving through the publication process ('Publication lags' in Supplementary Note 3). We further repeated our analyses by controlling ex post funding status for narrow wins and near misses ('Robustness for ex post funding status' in Supplementary Note 3). We also tried several name disambiguation methods and repeated our analyses ('Author name disambiguation' in Supplementary Note 1). Amid all variations, the conclusions remain the same.

The performance advantage of the near misses is particularly surprising given that the narrow wins, by construction, had an initial NIH funding advantage immediately after treatment. Given that funding from the NIH can be an important means to augmenting scientific production[35,38,39,52,56–58], we investigate funding dynamics for the near-miss and narrow-win groups over the following ten-year period. We find that the near-miss group naturally received significantly less NIH funding in the first five years following treatment, averaging $0.29 million less per person (Fig. 2d, $t$-test $p$-value $< 0.05$, Cohen's $d = 0.28$), which is consistent with prior studies[8,52,59]. Yet the funding difference between the two groups disappeared in the second five-year period (Fig. 2d, $t$-test $p$-value $> 0.1$, Cohen's $d = 0.02$). Although the NIH is the world's largest funder for biomedical research, near misses might have obtained more funding elsewhere (see 'Additional funding by near misses' in Supplementary Note 3 for details). To test this hypothesis, we further collected individual grant histories for PIs in our sample from the Dimensions data, allowing us to calculate the total funding support from agencies worldwide beyond NIH. We first measured the total funding support from the U.S. National Science Foundation (NSF) received by individuals with the same name in the same period, finding narrow wins obtained significantly more NSF funding within 5 years after treatment. We further calculated the total funding support from agencies other than the NIH or NSF, finding that near misses did not acquire more funding than narrow wins. We also manually checked acknowledgment statements within a fraction of papers published by the two groups, finding again the same conclusion.

Together, these results demonstrate that over the course of ten years, near misses had fewer initial grants from the NIH and NSF. Yet they ultimately published as many papers and, most surprisingly, produced work that garnered substantially higher impacts than their narrow-win counterparts.

Is the uncovered difference in outcomes causally attributable to the early-career setback? Or, could it be explained by other alternative forces? Indeed, there might still exist observable or otherwise unobserved factors that affect funding success near the threshold (e.g., individual characteristics[60], fields of study, personality traits, etc.), which might also drive future career outcomes. To rule out alternative explanations, we leverage two additional inference techniques, Coarsened Exact Matching (CEM)[61,62] and fuzzy Regression Discontinuity (RD)[63,64]. We first matched near misses and narrow wins with respect to a wide range of observable characteristics (see Methods section for further description of CEM), and find that after matching, near misses still outperformed narrow wins in terms of both hit papers (16.4% for near misses, 14.0% for narrow wins, $\chi^2$-test $p$-value $< 0.001$, odds ratio $= 1.20$) and average citations per paper (30.8 for near misses and 27.7 for narrow wins, $t$-test $p$-value $< 0.001$, Cohen's $d = 0.05$, see 'Matching strategy and additional results in the RD regression' in Supplementary Note 3 for details). While matching can only eliminate potential observable features, we further mitigate the effect of other observable and unobservable influences using the RD analysis. Specifically, we use an indicator for the score being above or below the funding threshold as an instrumental variable (IV), rather than the actual funding outcome itself, to predict future career outcomes (see Methods section). The RD approach helps us rule out unobserved influences on funding outcome or any otherwise unobserved individual characteristics that differ smoothly with the score[63,64], allowing us to further establish a causal link between early-career near miss and future career impact. By accounting for any potential confounding factors, our RD estimates indicate that one early-career near miss increases the probability of publishing a hit paper in the next 10 years by 6.1% ($p$-value $= 0.041$), and the average citations per paper by 34% (9.67 citations in 5 years, $p$-value $= 0.046$) (see Methods section). The RD analyses help establish the causal interpretation of our results, and the agreement in results across all the methods further demonstrates the robustness of our findings.

These results document that, despite an early setback, near misses outperformed narrow wins over the longer run, conditional on remaining active in the NIH system. This finding itself has a striking implication. Indeed, take two researchers who are seeking to continue their careers in science. While both near-miss and narrow-win applicants published high-impact papers at a higher rate than their contemporary peers, comparing between the two groups, it is the ones who failed that are more likely to write a high-impact paper in the future.

To conceptualize this finding, consider two hypotheses. The first is a screening hypothesis, where the population of survivors among the near-miss group may have fixed, advantageous characteristics. Second, the result is consistent with failure itself teaching valuable lessons or strengthening resolve. To help unpack the findings, we examine differential survival rates between two samples and further ask whether the screening hypothesis alone may be sufficient to explain the observed difference in outcomes.

**Screening hypothesis**. We first investigate attrition rates by studying the percentage of the initial PIs who remained active in the NIH system and find that the attrition rate of the two groups differed significantly (Fig. 3a). In the year immediately following treatment, the near-miss group had 11.2% fewer active PIs than the narrow-win group ($\chi^2$-test, $p$-value $< 0.001$). This difference is not simply because narrow wins received an initial grant. Indeed, the gap persisted and extended beyond the first five years, remaining at 11.8% in year seven ($\chi^2$-test, $p$-value $= 0.002$), followed by a drop afterwards. The RD analysis indicates that an early-career near miss on average led to a 12.6% chance of disappearing permanently from the NIH system over the next ten years (see Methods section). These results thus highlight the fragility of a junior scientific career, with one early near miss being associated with significantly higher attrition from the NIH system, despite the fact that to become an NIH PI, one had to go

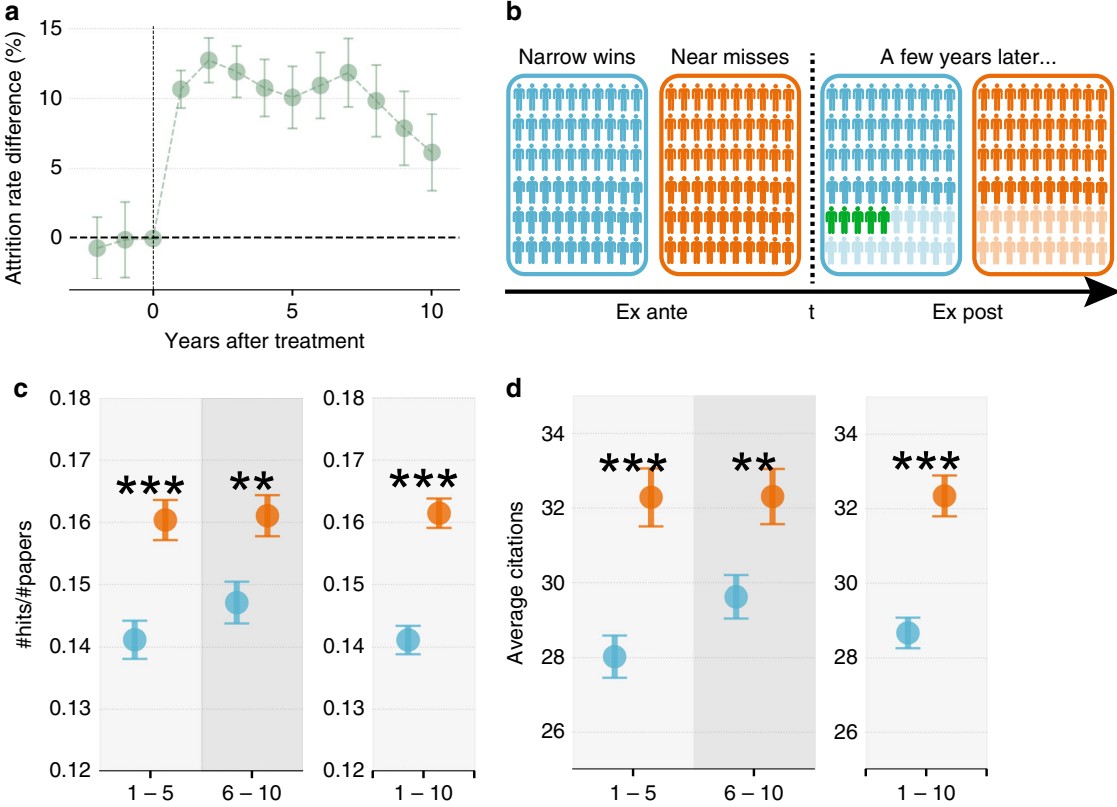

**Fig. 3** Testing the screening hypothesis with a conservative removal procedure. **a** Attrition rate difference between the near-miss and narrow-win group (near misses minus narrow wins). We measure the percentage of PIs remained in each of the two groups, and calculate their difference in each of the ten years after treatment. **b** An illustration of the conservative removal procedure. To test if the observed performance difference can be accounted for by the population difference, we performed a conservative estimation by removing PIs who published the fewest hit papers but with the most publications from the narrow-win group (blue), such that after removal (green) the two groups have the same fractions of PIs remaining. After removal, the near-miss group still outperformed the narrow-win group in terms of the probability of producing a hit paper ($\chi^2$ test $p$-value < 0.001, odds ratio = 1.17) (**c**), or the average citations of papers ($t$-test $p$-value < 0.001, Cohen's $d$ = 0.06) (**d**). The results shown in **c**–**d** suggest that while the performance of narrow wins indeed improved following the conservative removal procedure, the screening hypothesis alone cannot account for the uncovered performance gap. ***$p$ < 0.001, **$p$ < 0.05, *$p$ < 0.1; Error bars represent the standard error of the mean

through years of training with a demonstrated track record of research. Notwithstanding the evidence that PhDs who left science are disproportionally employed at large, high-wage establishments[65], Fig. 3a documents differential survivorship between narrow wins and near misses, which raises the important next question: Could screening alone account for the observed performance advantage?

To understand the nature of the potential screening effect, we first test its underlying assumption by comparing pre-treatment characteristics of near misses and narrow wins who remained ex post, finding a lack of difference between these two groups in any observable dimension ex ante (Supplementary Fig. 29a), which suggests the screening effect, if any, may be modest ('On the screening mechanism' in Supplementary Note 3). To further examine potential screening effects, we removed PIs from narrow wins, such that the attrition rate following removal is the same between the two groups (Fig. 3b). We performed a conservative estimation by removing PIs from narrow wins who, ex post, published the fewest hit papers but had the most publications. In other words, we created a subpopulation of narrow wins that had the same attrition rate as the near misses but are aided by an artificial upward adjustment to their hit probabilities ('On the screening mechanism' in Supplementary Note 3). We find that, while the performance of narrow wins improves by construction following this conservative removal procedure, the improvement

is not sufficient to account for the observed performance gap. Indeed, in terms of the probability of producing a hit paper, or the average citations per paper, near misses still outperformed narrow wins (Fig. 3c, d). The matching and the RD yield consistent conclusions ('Matching strategy and additional results in the RD regression' in Supplementary Note 3). Together, these results demonstrate that the screening effect may have played a role, but it appears insufficient to entirely account for the observed difference between near misses and narrow wins.

We clarify these results further on several dimensions. To understand if the average improvement of the near misses masks heterogeneous responses, we measured the coefficient of variation for citations, finding a lack of difference between the two groups, suggesting a homogeneous improvement within the group ('Variance and outliers' in Supplementary Note 3, Supplementary Fig. 23). We also compared the median citations to eliminate the role of outliers, yielding the same conclusion ('Variance and outliers' in Supplementary Note 3). To rule out collaboration effects, whereby early-career setbacks might lead junior scientists to seek out advantageous collaborations, we restricted our analyses to lead-author publications only, and controlled for the status of their collaborators, yielding the same conclusions (Supplementary Figs. 13, 27). Further, to check that the uncovered performance gap is not simply because narrow wins became worse, we selected a group of clear winners whose scores

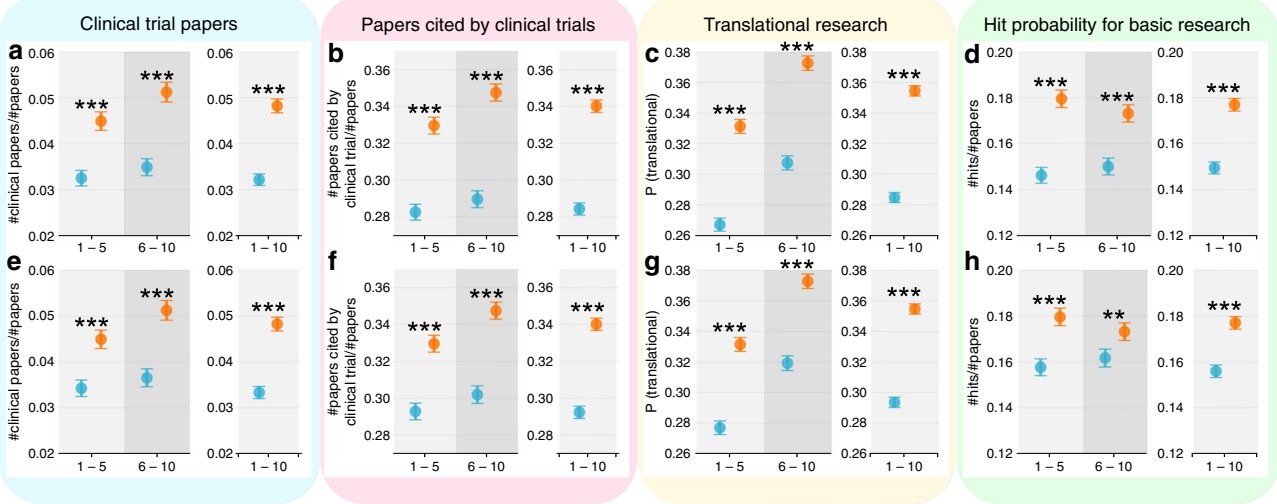

**Fig. 4** Near misses (orange) outperformed narrow wins (blue) in both basic and translational science. Here, we considered three measures probing the clinical relevance of their research: (1) clinical trial papers in the PubMed dataset (direct contribution to clinical translation); (2) papers cited by at least one clinical trial paper (indirect contribution to clinical translation); (3) papers with potential to become translational research. Specifically, the approximate potential to translate (APT) score was used to identify early signatures of bench-to-bedside translation. The score is estimated using machine learning combining features such as MeSH terms, disease, therapies, chemical/drug, and citation rates. **a** Near misses outperformed narrow wins in terms of the probability of producing clinical trial papers in the next 1–5 years, and 6–10 years; **b** The same as a but for papers cited by clinical trials in the future; **c** The same as a but for papers with potential to be a translational research; **d** Hit paper probability by considering non-clinical trial papers only. **e–h** The same as **a–d** but for the conservative removal (following the same procedure in Fig. 3b). ***$p < 0.001$, **$p < 0.05$, *$p < 0.1$; Error bars represent the standard error of the mean

were further removed from the funding threshold. We find, as expected, that this group of PIs performed substantially better than the near-miss group prior to treatment. Yet, in the ten years afterwards, they show a similar performance as the near-miss group ('Was it because narrow wins became worse?' in Supplementary Note 3), indicating that near misses performed at a comparable level as the group that appeared demonstrably better than them initially. To test the hypothesis that narrow wins were committed to initially proposed ideas, we compared articles by narrow wins published in 5 years after treatment with those published between 6 to 10 years. We find no statistically significant improvement for narrow wins in terms of probability to publish hit papers ($\chi^2$-test $p$-value $> 0.1$) or normalized citations ($t$-test $p$-value $> 0.1$) ('Was it because narrow wins became worse?' in Supplementary Note 3). We also controlled for fixed effects categorizing PIs' prior NIH experience, recovering the same conclusions (Supplementary Fig. 31). We also repeated all our analyses by varying our definition of active PIs by focusing on publishing scientists only ('Different definitions of active PIs' in Supplementary Note 3, Supplementary Fig. 25), and the definition of pay lines by using the NIH percentile score instead of priority score (Supplementary Fig. 26). Amid all variations, our findings remain the same.

**Beyond citations**. While citations and their variants have been used extensively to quantify career outcomes[7,41,45,66–69], they may represent an imperfect or limited proxy for measuring output, prompting us to ask if the observed effect of early-career setback extends beyond citation measures. To this end, we used additional datasets to calculate three indicators probing the clinical relevance of the works. These measures are: (1) whether a paper is a clinical trial publication (direct contribution to clinical translation); (2) whether a paper has been cited by at least one clinical trial publication (indirect contribution to clinical translation), and (3) whether a paper has potential to become translational research[70] (potential for translation). We compared works

produced by near misses and narrow wins over the ten-year period, finding that across all three translational dimensions, near misses systematically outperformed narrow wins. Specifically, near misses were 50% more likely to publish a clinical trial paper compared with narrow wins (4.8% for near misses, 3.2% for narrow wins, $\chi^2$-test $p$-value $< 0.001$, odds ratio $= 1.53$, Fig. 4a), and their overall publications were 19.6% more likely to be cited by clinical trials (34% for near misses, 28.4% for narrow wins, $\chi^2$-test $p$-value $< 0.001$, odds ratio $= 1.30$, Fig. 4b), and are 24.5% higher in their potential for bench-to-bedside translation (35.4% for near misses, 28.4% for narrow wins, $\chi^2$-test $p$-value $< 0.001$, odds ratio $= 1.38$, Fig. 4c). We also find that, all these conclusions remain the same after conducting the conservative removal procedure as described in Fig. 3 (Fig. 4e–g). Finally, to test if the tendency toward clinical research can by itself account for the observed citation difference between near misses and narrow wins, we separated their publications into clinical and non-clinical papers, finding that within non-clinical papers, near misses again outperformed narrow wins (Fig. 4d, h). We repeated all our analyses using the RD approach, recovering broadly consistent conclusions ('Matching strategy and additional results in the RD regression' in Supplementary Note 3). Together, the results shown in Fig. 4 suggest that the uncovered effect of early-career setback goes beyond citation measures, with near misses outperforming narrow wins in both basic and translational science.

## Discussion
Overall, these results document that an early-career setback has powerful, opposing effects, hurting some careers, but also, quite surprisingly, strengthening outcomes for others. As such, these findings show that prior setback can indeed be a mark of future success. Screening effects may partly be responsible yet appeared insufficient to explain the magnitude of the observed effects, supporting the idea that failure may teach valuable lessons[19–21,25].

The uncovered effects may operate according to multi-dimensional mechanisms. We explored ten different observable dimensions, including shifts in intellectual direction and leadership, institutional locus, and collaboration patterns (see Methods section). We find only a suggestive tendency for near-miss scientists to publish in hot topic areas following treatment, although accounting for this factor did not reduce the observed performance gap (Supplementary Fig. 28). More generally, these numerous observable features considered do not account alone or collectively for the performance change, suggesting that unobservable dimensions may play a role, including effort, signaling, and grit factors following setbacks[21,23] or sacred sparks[71]. Crucially, the empirical findings and conclusions reported in the paper hold the same, regardless of the underlying processes. Indeed, while most empirical and theoretical evidence in science thus far documents that individuals benefit tremendously from success[1,2,4–8,29,52,59], our results offer among the first empirical evidence showing that some individuals can also benefit from setbacks, which may have broad implications for both individual investigators and institutions that aim to support and nurture them.

The design of our study necessitates the focus on near-miss individuals among all others who had setbacks[72,73]. As with any RD analysis, the effect pertains to the specific population on which the experiment was conducted. While the RD approach allows us to expand our sample to a wider range of setbacks by controlling for evaluation scores, yielding the same conclusions (see Methods section), to what degree our findings may generalize substantially beyond near misses is a question we cannot yet answer conclusively. Moreover, the opposing effects of early setbacks also suggest there may exist population heterogeneities in responses that are worth exploring further. Who tends to be the most vulnerable, and who the most resilient? Quantitative answers to these questions may be crucial for the interpretation of our insights to inform policies and intervention strategies for building a robust scientific workforce[74].

Moreover, our analyses estimate the net advantage of near misses over narrow wins, which is only detectable if the gross advantage of early-career failure outweighs any benefits conferred by success. Given the widespread, convincing evidence supporting the validity of the Matthew effect in science[1,3–9,29,36] and beyond[3,10–13], where past success begets future success, these results suggest that powerful, offsetting mechanisms may be at work. This implies that, in areas where the Matthew effect operates less, the net advantage of failure may be more pronounced, suggesting that other domains provide important additional avenues for future work.

Finally, note that our results do not imply that one should strategically put roadblocks in the way of junior scientists, as the precondition of becoming stronger is to not be killed in the first place. The findings do suggest, however, that for those who persevere, early failure should not be taken as a negative signal—but rather the opposite, in line with Shinya Yamanaka's advice to young scientists, after winning the Nobel prize for the discovery of iPS cells, "I can see any failure as a chance."

## Methods

**Testing various generative processes**. While the broad idea of a setback-driven boost may take many forms, several such mechanisms may be detectable from data in our context. For example, (A) did early-career setbacks propel persistent junior scientists to attempt more novel research, whereas narrow wins were bound to their original ideas? (B) Did early-career setbacks lead junior scientists to seek out advantageous collaborations? Indeed, as teams are increasingly responsible for producing high-impact work[33,75], the observed performance gap might reflect collaborations, whereby near misses more frequently teamed up with higher-impact scientists and/or published fewer lead-author publications than their narrow-win counterparts. Alternatively, the uncovered difference might reflect an

observable personal change process in terms of intellectual or physical mobility, as captured by research direction shifts (hypothesis C)[76] or changing institutions (hypothesis D)[77,78]. We tested hypotheses A–D from our data, finding that there is only a suggestive tendency for near-miss scientists to publish in hot topic areas following treatment, although accounting for this finding does not reduce the observed performance gap (Supplementary Fig. 28, 'Combining hypotheses A-D' in Supplementary Note 4). However, none of the hypotheses alone can fully explain the observed performance gap between near misses and narrow wins (see Supplementary Note 4 for details, Supplementary Fig. 28); nor do these hypotheses combine to explain the findings when we control for all the factors outlined in hypotheses A–D together ('Combining hypotheses A-D' in Supplementary Note 4). Overall, investigating these many dimensions narrows the potential interpretations of our results while further suggesting that additional sub-processes may be at work, including effort or grit factors following setbacks[21,23], which are difficult to observe directly from data and provide areas for future research. Crucially, the empirical findings reported in the paper hold the same, net of potentially many underlying processes.

**Econometric model specification and estimation procedures**. A possible concern with the comparison between the near misses and narrow wins near the threshold is endogeneity;[52] i.e., there might be other factors that influence both the funding decision and future career outcome. The finding that observable features of the two groups are statistically indistinguishable prior to the funding decisions helps diminish this concern. Further, were there some unobserved factor determining outcomes, such a factor would need to both advantage the narrow wins in getting initial funding yet disadvantage them over the longer run, which may be implausible. Nevertheless, to fully eliminate such endogeneity concerns, one needs to employ a causal inference framework called fuzzy regression discontinuity design (RD)[63]. In this section, we describe in detail our econometric model specification and estimation procedures.

The key idea of RD is that if decision rules create a jump in the likelihood of treatment, often at an arbitrary threshold, we can exploit this local discontinuity as an exogenous variable to predict outcome, instead of using the specific, realized treatment outcome, which could be influenced by endogenous factors. Such RD approaches have been widely used in the fields of education, labor, political economy, health, crime, and environmental studies[64], in addition to prior research on funding data[8,52,58,79].

In our setting, the likelihood of treatment (i.e., receiving funding) is largely determined by the score of each proposal[52,79]. The highly nonlinear relationship between the evaluation score and funding success (Fig. 1a) makes our fuzzy regression discontinuity design feasible. More specifically, we can treat whether scores fell just above or below the cutoff as an instrument to predict funding, and then use the predicted funding outcome, rather than the actual funding outcome, to predict future career outcome. The logic of this procedure is the following. Both the actual funding outcome and future career outcome can be affected by observable or unobservable factors. But since whether or not the score of the proposal is above or below the funding threshold is not influenced by any endogenous factors, if that as an instrumental variable can predict future career outcome, then it means there must exist a link from funding outcome to future career outcome, because the only way for the instrumental variable to influence future career outcome is through the funding outcome. Detailed causality diagram is shown in Supplementary Fig. 6. Another advantage of using the instrumental variable approach is to allow us to control for the score itself (the so-called running variable in RD), and thus control for any distinctions in applications that vary smoothly with the score.

More specifically, we estimate the causal effect of early-career setbacks on future career outcomes using two-stage least squares regression (2SLS): in the first stage, the instrumental variable (being above or below the score threshold) is used to predict the funding outcome; in the second stage future career outcomes are regressed on the predicted funding outcome from the first stage. As illustrated in Supplementary Fig. 6, the fuzzy RD approach eliminates any potential unobservable factors (e.g., novelty bias, hot topic, grit personality, perseverance, and so on) that might affect both the funding and career outcomes. This means, any significant results obtained from this approach can be interpreted as causal relationships.

More specifically, given the jump in the probability of receiving the funding at $s_0$ (i.e., normalized score 0), we have

$$P\left(F_j = 1 | s_j\right) = \begin{cases} g_1\left(s_j\right) & \text{if } s_j \geq s_0, \\ g_2\left(s_j\right) & \text{if } s_j < s_0 \end{cases}, \qquad (1)$$

where $s_j$ is the running variable, i.e., normalized score for proposal $j$, $F_j$ is the funding decision outcome, and $g_1\left(s_j\right) \neq g_2(s_j)$ at $s_0$. The probability to receive treatment is

$$E\left[F_j | s_j\right] = P\left(F_j = 1 | s_j\right) = g_2\left(s_j\right) + \left[g_1\left(s_j\right) - g_2\left(s_j\right)\right] z_j, \qquad (2)$$

where $z_j = 1$, if $s_j \geq s_0$ and $z_j = 0$ otherwise. Since $s_0$ is the normalized score with a funding threshold that divides the proposals at an arbitrary point, the dummy

variable $z_j$ is uncorrelated with any observed or unobserved factors[52]. To this end, we treat $z_j$ as the instrument variable and employ a simple two-stage least square (2SLS) regression estimation. Let $y_{it}$ be some scientific outcome of individual $i$ during time period $t$, i.e., probability to publish hit papers, number of publications, or number of hit papers. We conduct the estimation as follows:

$$1^{st} stage: F_j = \alpha_0 + \alpha_1 s_j + \alpha_2 s_j^2 + \cdots + \alpha_p s_j^p + \pi_i z_j + \theta \mathbf{X}_{i,pre} + \mu_t + \eta_n + \eta_j, \quad (3)$$

$$2^{nd} stage: y_{it} = \beta_0 + \beta_1 s_j + \beta_2 s_j^2 + \cdots + \beta_p s_j^p + \lambda \hat{F}_j + \gamma \mathbf{X}_{i,pre} + \mu_t' + \eta_n' + \in_i, \quad (4)$$

where $\mathbf{X}_{i,pre}$ is the prior performance of researcher $i$, i.e., prior number of publications, prior number of hit papers; $\mu_t$ and $\mu_t'$ are grant time fixed effects for application year, $\eta_n$ and $\eta_n'$ are NIH institution fixed effects, $p$ is the order of the polynomial control of the running variable, and $\eta_j$ and $\in_i$ are error terms from the first and second stage, respectively. Moreover, $\hat{F}_j$ is the predicted values from the first stage, which are uncorrelated with the error term $\in_i$, and $\lambda$ is the causal effect of near miss on future career outcomes. For a large sample that span a significant fraction of PIs around 0, we need careful controls of priority score. In the following analyses, we added the linear control of score when considering ±10 discontinuity sample ($p = 1$), and 3rd order polynomial control for ±25 discontinuity sample ($p = 3$). Finally, we eliminate these polynomial controls as we restrict the sample to the very narrow region around the discontinuity point[63] (in our setting, ±5 discontinuity sample).

The credibility of these estimates hinges on the assumption of the lack of prior knowledge of the cutoff, $s_0$, so that individual scientists cannot precisely manipulate the score to be above or below the threshold. This assumption is valid in our setting, because the scores are given by external reviewers, and cannot be determined precisely by the applicants. To offer quantitative support for the validity of our approach, we run the McCrary test[80] to check if there is any density discontinuity of the running variable near the cutoff, and find that the running variable does not show significant density discontinuity at the cutoff (bias = −0.11, and the standard error = 0.076). Together, these results validate the key assumptions of the fuzzy RD approach.

To understand the effect of an early-career near miss using this approach, we first calculate the effect of near misses for active PIs. Using the sample whose scores fell within −5 and 5 points of the funding threshold, we find that a single near miss increased the probability to publish a hit paper by 6.1% in the next 10 years (Supplementary Fig. 7a), which is statistically significant ($p$-value < 0.05). The average citations gained by the near-miss group is 9.67 more than the narrow-win group (Supplementary Fig. 7b, $p$-value < 0.05). By focusing on the number of hit papers in the next 10 years after treatment, we again find significant difference: near-miss applicants publish 3.6 more hit papers compared with narrow-win applicants (Supplementary Fig. 7c, $p$-value 0.098). All these results are consistent with when we expand the sample size to incorporate wider score bands and control for the running variable (Supplementary Fig. 7a-c).

For our test of the screening mechanism, we employ a conservative removal method as described in the main text (Fig. 3b) and redo the entire regression analysis. We recover again a significant effect of early-career setback on the probability to publish hit papers and average citations (Supplementary Fig. 7d, e). For hits per capita, we find the effect of the same direction, and the insignificant differences are likely due to a reduced sample size, offering suggestive evidence for the effect (Supplementary Fig. 7f). Finally, in order to test the robustness of the regression results, we further controlled other covariates including publication year, PI gender, PI race, institution reputation as measured by the number of successful R01 awards in the same period, and PIs' prior NIH experience. We recovered the same results (Supplementary Fig. 17).

**Coarsened exact matching**. To further eliminate the effect of observable factors and consolidate the robustness of the results, we employed the state-of-art method, i.e., Coarsened Exact Matching (CEM)[61]. The matching strategy further ensures the similarity between narrow wins and near misses ex ante. The CEM algorithm involves three steps:

1. Temporarily coarsen each control variable $\mathbf{X}$ for the purposes of matching;
2. Sort all observations into strata, each of which has the same values of the coarsened $\mathbf{X}$.
3. Prune from the data set the units in any stratum that do not include at least one treated and one control unit.

Following the algorithm, we use a set of ex ante features to control for individual grant experiences, scientific achievements, demographic features, and academic environments; these features include the number of prior R01 applications, number of hit papers published within three years prior to treatment, PI gender, ethnicity, reputation of the applicant' institution as matching covariates. In total, we matched 475 of near misses out of 623; and among all 561 narrow wins, we can match 453. We then repeated our analyses by comparing career outcomes of matched near misses and narrow wins in the subsequent ten-year period after the treatment. We find near misses have 16.4% chances to publish hit papers, while for narrow wins this number is 14.0% ($\chi^2$-test $p$-value < 0.001, odds ratio = 1.20, Supplementary Fig. 21a). For the average citations within 5 years after publication, we find near misses outperform narrow wins by a factor of 10.0% (30.8 for near

misses and 27.7 for narrow wins, $t$-test $p$-value < 0.001, Cohen's $d = 0.05$, Supplementary Fig. 21b). Also, there is no statistical significant difference between near misses and narrow wins in terms of number of publications. Finally, the results are robust after conducting the conservative removal ('Matching strategy and additional results in the RD regression' in Supplementary Note 3, Supplementary Fig. 21d-f).

**Reporting summary**. Further information on research design is available in the Nature Research Reporting Summary linked to this article.

## Data availability

This paper makes use of restricted access data from the National Institutes of Health, protected by the Privacy Act of 1974 as amended (5 U.S.C. 552a). De-identified data necessary to reproduce all plots and statistical analyses are available at https://yang-wangnu.github.io/setback/. Those wishing to access the raw data may apply for access following the procedures outlined in the NIH Data Access Policy document available at http://report.nih.gov/pdf/DataAccessPolicy.pdf. The Web of Science data are available via Clarivate Analytics. The PubMed data are publicly available as discussed in the supplementary information. We obtained the Dimensions data from https://www.dimensions.ai.

## Code availability

Code is available at https://yang-wangnu.github.io/setback/.

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

## Acknowledgements

The authors thank A.-L. Barabási, J. Chown, J. Evans, E. Finkel, V. Medvec, J. Loscalzo, W. Ocasio, P. Stephan, B. Uzzi, Y. Yin, and all members of Northwestern Institute on Complex Systems (NICO) for invaluable comments. This work is supported by the Air Force Office of Scientific Research under award number FA9550-15-1-0162, FA9550-17-1-0089, and FA9550-19-1-0354, Northwestern University's Data Science Initiative, the National Science Foundation grant SBE 1829344, and Alfred P. Sloan Foundation Award G-2015-14014. This work does not reflect the position of NIH.

## Author contributions

D.W. and B.F.J. designed research; Y.W. performed research, analyzed data; all authors wrote and edited the paper.

## Additional information

**Competing interests:** Y.W. and D.W. serve as special volunteers (unpaid) to the NIH. B. F.J. declares no competing interests.

