## [Peer Review File · Nature Communications]

Reviewers' comments:

Reviewer #1 (Remarks to the Author):

What is the impact of early-career setbacks on future success in science? The authors find that, not only do these early setbacks produce equally impactful scientists, but that early setbacks produce more impactful scientists (as measured by citations) than their early-career setforwards. The authors use a combination of NIH grant ratings and bibliographic databases. The result is intriguing and music to the ears of all those receiving setbacks. Others have investigated this question, but the paper here is one of the more extensively investigated in the literature. The authors are careful in testing and eliminating alternative hypothesis (e.g., screening mechanisms). Overall, I think the paper is an important contribution to the field. However, I still have some questions that I hope can be addressed to further convince me of the results.

First, a small clarification for the non-economists reading the paper: The data used for the analysis was impressive (all grants from 1990 - 2005), but as with any RD design, ultimately the analysis narrowed to a relatively small subset of researchers. This needs to be clearer in the paper. The analysis set was narrowed to 561 near-win authors and 623 near-miss authors, but this was buried in a figure caption. Most readers will not be familiar with RDD. This changes how the results are interpreted. In the paper, the authors boast throughout that they "leveraged a unique dataset, containing all R01 grant applications ever submitted to the NIH." That is true, but relatively few authors were used in making judgments about near-misses. I find this a bit misleading.

The narrowing of the data set creates a manageable set for investigating other possible explanations. For example, the authors correctly note that near-miss researchers may have received funding from other outside sources. Did the authors look at the CVs or personal websites to see if these authors received other funding? It could be that some of the near-misses were not near-misses at all. They may have been just fine on funding. It would help to know better what these 623 researchers did after missing the first NIH grant.

How do the authors account for publication lags? Could it be that many of the papers published after the near miss were written before the near-miss but were moving through the publication process? One could also see where the hit papers occurred in the ten period. If they are occurring shortly after the near miss, then it may be more important to look into this. One could look at preprints or "date of acceptance" versus print.

When setting up the RD design, were there disproportionately more papers at an early/later time period for the near-miss/near-win groups? Later time period papers likely receive more citations because there are more papers that are citing. One could set up the RD design to account for this, but it wasn't clear if the authors did this and accounted for the time period effect.

How sensitive are the results to perturbations in fields? Identifying fields is difficult. They can change dramatically depending on which classifications are used. The authors have explored perturbations for other factors such as the percentage of hit papers, have the authors perturbed the field schemas for setting up the RDD.

For the near-misses, how long does it take to receive the next funding grant? The authors provided some information on this but I wonder if this creates the need for subsetting RDD. For example, matching authors (both near-win and near-miss authors) that say receive a grant one year after the near-miss/near-win. This would align those grantees that receive their follow-up funding.

Grant scores are a blunt instrument for accessing why a grant was denied. Do the authors have any other information for those scores? I see so many issues that are hidden under the scores. For example, it could be that a grant was denied, not for quality issues, but for scoping issues. That

gets lost in this data. It is also where I think real mechanisms reside for explaining the near misses.

Small thing: The “what doesn’t kill you, makes you stronger” quotes are a little overdone. I think the point is made strongly enough. You could probably remove one or more of the quotes.

Reviewer #2 (Remarks to the Author):

The authors empirically compared two groups of applicants applying for U.S. National Institutes of Health (NIH) R01 grants: “near-miss” and “near-win”. The study is interesting and methodologically sound. However, it seems that the authors have not searched previous peer review literature systematically. Studies with similar results (similar ex-post performance of best-rejected and approved applicants) have been published earlier (e.g., Melin & Danell, 2006). An overview of studies can be found in Bornmann (2011). The following points should be considered in a revision of the manuscript:

Paper

Page 2: Another possible explanation for the results might be that near-miss applicants searched for signs of success (e.g., publications in high-impact journals) more intensively than near-wins, since near-wins are able to demonstrate their success by receiving the R01 grants. We possibly know this effect from statements by Nobel Laureates in which they refrain from publishing in high-impact journals. They can say that: since they received the Nobel Prize, it is not necessary for their career to focus on signs of academic success besides the prize.

Page 2: One should consider in the interpretation of the results that the authors analyzed a highly selective group of scientists. Setbacks might be goosing only for people who know their potential (near-miss applicants, because they already had success otherwise), but might have different effects on other scientists (e.g., disillusioning or frustrating).

Page 3: There are three theories available to explain success or failure in scientific careers: the “sacred spark” theory (Cole & Cole, 1973), the “accumulative advantage” theory (Merton, 1968), and the “utility maximizing theory.” According to the last theory, “all researchers choose to reduce their research efforts over time because they think other tasks may be more advantageous” (Kwiek, 2015). All these theories should be considered in the manuscript under review.

Page 3: The authors have an interesting study design which can be improved, however, in my opinion: (1) the authors included mostly performance indicators in the comparison of near-miss and near-win groups and only a few demographics. I recommend to include a broader set of variables considered in the comparison (nationality, reputation of previous institutions, number of children etc.). Factors should be considered, which do not only have a possible effect on peer review decisions (thus, take a look at the corresponding bias literature, see Bornmann, 2011), but also on careers of scientists. (2) The design is not convincing: The authors should ensure that both groups (near-miss and near-win) are really similar. In order to investigate in a quasi-experimental evaluation study the outcome of “social intervention programs”, Rossi, Freeman, and Lipsey (1999) favor the “comparison group design”: “Targets receiving the program are compared with some selected group of targets or potential targets who do not receive the program” (p. 310). They recommend to generate pairs of two people (so-called “artificial twins”) for a treatment group (people receiving the program) and a control group (people not receiving the program). Both groups should be as similar as possible. For example, Baker, Robertson, and Toguchi (1996) selected for APRA fellowship holders “a matched sample of non-APRA holders” (p. 24).

Page 6: The authors write: “All else equal, except that one has had an early funding failure and

the other an early funding success, it is the one who failed that is expected to write higher-impact papers in the future." In my opinion, this conclusion is too strongly formulated. The probability is higher, but highly cited papers can be expected from near-win applicants too (with higher probability than expected).

Page 9: The authors write: "they represent an imperfect proxy at best to measure scientific outputs, suggesting that future research may fruitfully explore measurements that extend beyond citation-based measures to examine broader career outcomes". In my opinion, this should not only be a point in the discussion section, but should be done by the authors in their study. Please use other indicators different from bibliometrics to validate your results.

Figure 1: The case numbers are mentioned below the figure (561 near-win and 623 near-miss applicants), but not in the text. However, this is an important information.

Supporting Information

Page 3: It is no longer Thomson Reuters, but Clarivate Analytics.

Page 4: How are fields defined? The best field categorization scheme which can be used for comparison of citation impact might be MeSH terms (Bornmann, Marx, & Barth, 2013). Bibliometric studies have pointed out several weaknesses of using journal sets as field categorization scheme.

Pages 5 and 6: Tekles and Bornmann (2019) investigated several methods for author disambiguation. Their analyses show that the method published by Caron and van Eck (2014) performs best. The authors of the manuscript under review should orient towards such analyses to decide which disambiguation method is used in their study. Furthermore, a sample of PIs should be drawn and manually checked whether the disambiguation method works satisfactorily or not.

Pages 11 and 12: It is not clear how fields have been defined.

Page 12: The authors used the RCR indicator in their study: The indicator has been critically assessed in bibliometrics (Waltman, 2015).

References

- Baker, M., Robertson, F., & Toguchi, H. (1996). The Australian postgraduate research award scheme: An evaluation of the 1990 cohort. Canberra, Australia: Higher Education Division.
- Bornmann, L. (2011). Scientific peer review. *Annual Review of Information Science and Technology*, 45, 199-245.
- Bornmann, L., Marx, W., & Barth, A. (2013). The normalization of citation counts based on classification systems. *Publications*, 1(2), 78-86.
- Caron, E., & van Eck, N. J. (2014). Large scale author name disambiguation using rule-based scoring and clustering. In P. Wouters (Ed.), *Proceedings of the science and technology indicators conference 2014 Leiden "context counts: Pathways to master big and little data"* (pp. 79-86). Leiden, the Netherlands: University of Leiden.
- Cole, J. R., & Cole, S. (1973). *Social stratification in science*. Chicago, MA, USA: The University of Chicago Press.
- Kwiek, M. (2015). The European research elite: A cross-national study of highly productive academics in 11 countries. *Higher Education*, 71(3), 379-397. doi: 10.1007/s10734-015-9910-x.
- Melin, G., & Danell, R. (2006). The top eight percent: Development of approved and rejected applicants for a prestigious grant in Sweden. *Science and Public Policy*, 33(10), 702-712.
- Merton, R. K. (1968). *Social theory and social structure*. New York, NY, USA: Free Press.
- Rossi, P. H., Freeman, H. E., & Lipsey, M. W. (1999). *Evaluation: A systematic approach*. London, UK: Sage Publications.
- Tekles, A., & Bornmann, L. (2019). Author name disambiguation of bibliometric data: A

comparison of several unsupervised approaches. Retrieved April 30, 2019, from <https://arxiv.org/abs/1904.12746>

Waltman, L. (2015). Nih's new citation metric: A step forward in quantifying scientific impact? Retrieved November 10, 2015, from <http://www.cwts.nl/blog?article=n-q2u294&title=nihs-new-citation-metric-a-step-forward-in-quantifying-scientific-impact#sthash.w1KC3A1O.dpuf>

Reviewer #3 (Remarks to the Author):

This paper studies the effect of early career setbacks (i.e. failure in NIH R01 grant applications) on career development and finds that "near misses" outperform "near wins," suggesting "what doesn't kill me makes me stronger." The finding is very novel and has important policy implications. The research is rigorously designed and executed, carefully ruling out various alternative explanations and implementing various robustness tests. The paper is definitely an important contribution and should be published. On the other hand, I also have some comments and suggestions.

Comparing "near miss" and "near win" and regression discontinuity design allows the authors to make strong causal claims. However, one important limitation is that it is only about the local effect around the cut-off point. In other words, the findings might not be generalizable to applicants who have a very high review score, namely we do not know whether declining their funding will actually help their performance. Furthermore, in this special context, we might expect that the local effect is very special. Specifically, there might be an alternative explanation: it might not be about setbacks but rather about "undeserved success and luck." The grant review process is supposed to screen out "unworthy" projects. Near misses and near wins are both "sub-optimal" projects. Failing the grant application allows the candidate to further develop their ideas or change to more promising ideas, while successful applicants commit to sub-optimal ideas and miss the opportunity of finding better directions. Therefore, it is actually about "underserved success" rather than setbacks. The robustness tests (S3.10) is related to this, but it is in line with this alternative explanation rather than rejecting it. The authors should address this alternative explanation.

One complementary analysis might be: using whether having a failed grant application before (or the number of failed attempts) as an independent variable, and then examine how it affects evaluation score, and conditional on having the same score, how does it affect further career development.

Point-By-Point Response

Referee #1

What is the impact of early-career setbacks on future success in science? The authors find that, not only do these early setbacks produce equally impactful scientists, but that early setbacks produce more impactful scientists (as measured by citations) than their early-career setforwards. The authors use a combination of NIH grant ratings and bibliographic databases. The result is intriguing and music to the ears of all those receiving setbacks. Others have investigated this question, but the paper here is one of the more extensively investigated in the literature. The authors are careful in testing and eliminating alternative hypothesis (e.g., screening mechanisms). Overall, I think the paper is an important contribution to the field. However, I still have some questions that I hope can be addressed to further convince me of the results.

Response: Thank you very much for these positive comments about our paper. Next, we offer a detailed point-by-point response to your insightful questions and suggestions.

1.1 *First, a small clarification for the non-economists reading the paper: The data used for the analysis was impressive (all grants from 1990 - 2005), but as with any RD design, ultimately the analysis narrowed to a relatively small subset of researchers. This needs to be clearer in the paper. The analysis set was narrowed to 561 near-win authors and 623 near-miss authors, but this was buried in a figure caption. Most readers will not be familiar with RDD. This changes how the results are interpreted. In the paper, the authors boast throughout that they “leveraged a unique dataset, containing all R01 grant applications ever submitted to the NIH.” That is true, but relatively few authors were used in making judgments about near-misses. I find this a bit misleading.*

Response: Thank you for this comment and suggestion. As with any RD analysis, we necessarily focused on a narrow region within our data (in our case, around the NIH payline) to allow for causal interpretation of our results. We fully agree that we should have made the point about sample size more clearly in the paper, especially given the broad potential readership for this paper.

Following your suggestion, we added a specific statement in the main text to draw readers’ attention to this fact, and mentioned explicitly the observation counts. This new text is as follows:

In total, our analyses yielded 561 near winners and 623 near misses around the payline.

1.2 *The narrowing of the data set creates a manageable set for investigating other possible explanations. For example, the authors correctly note that near-miss researchers may have received funding from other outside sources. Did the authors look at the CVs or personal websites*

to see if these authors received other funding? It could be that some of the near-misses were not near-misses at all. They may have been just fine on funding. It would help to know better what these 623 researchers did after missing the first NIH grant.

Response: Thank you very much for this helpful comment. We have undertaken several extensive investigations in response, which we describe in sequence.

First, following your suggestion, we went ahead to collect information for both near winners and near misses by searching for their CVs and personal websites online to see if near misses received more funding from agencies other than NIH. Our initial procedure was to search for the names and affiliation information for all PIs through google to see whether any homepage or CV appeared on the first page of search results. Among these searches, we found that 66% returned official homepages (i.e., university faculty pages that contain mostly biographic information only). However, we found that only 2.8% of these search results presented personal CVs.

Given this limited coverage, we further drilled down on 200 PIs for detailed screening, and investigated each individual manually and extensively. Specifically, we randomly selected 100 scientists from the near-win group, 100 from the near-miss group, and manually searched for their profiles (including lab websites, personal websites, personal CVs etc.). After an extensive search process, however, it proved difficult to find information about funding. In particular, across all these individuals, our efforts yielded 13 (of 100) near-miss cases with information about prior funding, and 10 (of 100) near-win cases with such information. Therefore, despite extensive effort, the results indicate that this approach provides very limited data and it is therefore unlikely to yield meaningful statistics to compare the two groups.

We realize that there have been very successful attempts at collecting CVs in the literature¹⁻³, and there are several reasons why it may be more difficult to collect this data in our case. For example, whereas having a website is a norm for computer scientists, it is less so for biomedical scientists. Also, while CVs tend to list publications, they are rather mixed when it comes to listing funding records.

Nevertheless, we fully agree with you that it would be great to further test alternative funding sources, and it would make the paper much stronger if we can. Therefore, although collecting CVs or personal websites didn't quite work out, to overcome this roadblock, we further tried *three* new approaches, including making use of a highly novel and authoritative dataset to help eliminate this concern. We next describe these approaches in detail:

Approach A: acquired a novel data capturing individual funding history:

Your comment prompted us to search for large-scale datasets that can capture individual grant records, which led us to discover a new data source that seems ideal for our purpose: the Dimensions data, a major new data product from Digital Science. This database specializes in collecting funding histories for each individual scientist by assembling and integrating all publicly available funding records from all agencies around the world. It contains more than 4.8 million funding records funded by 340 funding agencies from 40 countries and represents to our knowledge the most authoritative data source for this purpose.

To access the data required institutional agreements which contributed to the delay of our response, but we are very excited about this new data as we believe it is quite helpful for addressing questions about funding sources. Indeed, as we show next, having access to this data brings much clarity to this issue, and allowed us to draw more clear conclusions.

For each PI in our sample, we manually searched for scientists with the same name in the same period, and retrieved his/her grant history recorded in the data using the online interface on Dimensions. The results are as follows.

First, the results show that near winners received significantly more NIH funding within 5 years after treatment (t -test, p -value = 0.04), but not for year 6 to year 10 (p -value = 0.54). The results agree with the original results we reported in the manuscript (fig. 2d) where we use the NIH data directly, and suggest the validity of the Dimensions data. Second, near winners also obtained more NSF funding in the next five-year window (p -value = 0.02), but not for year 6 to year 10 (p -value = 0.73). The results so far indicate that it is the near-win group who received more funding within 5 years after treatment. We then measured the funding amount received by each PI from agencies other than NIH or NSF in the ten-year window following the treatment. We find the median funding amount per person for near misses between year 1 to year 5 is \$370,000, and \$610,000 for near winners, showing marginal statistical significant difference between the two group with near winners having slightly more funding on average (p -value = 0.07). We then measured the funding amount for the two groups between year 6 to year 10, obtaining \$670,000 and \$600,000 per person for near misses and near winners, respectively (p -value = 0.34). We also find near winners obtained slightly more funding when comparing the two groups between year 1 to year 10 (p -value = 0.08). Together these results demonstrate that in the ten-year period, near misses did not acquire more funding from agencies other than NIH or NSF, compared with near winners.

While the approach above offers large-scale, quantitative evidence for this question, in this revision, we further performed two additional approaches that offer further support, as follows.

Approach B: collecting funding acknowledgement from PubMed data:

We also looked into the PubMed acknowledgement data, which contains research grants supported by any agency of the United States Public Health Service from 1981 (e.g., NIH, CDC, FDA, etc.). After 2005, the dataset also includes grant information for many other US or non-US funding agencies and organizations. Though we do not have funding amount in this dataset, we find that there is a lack of difference between near misses and near winners in the number of agencies other than the NIH or NSF. Specifically, within 5 years after treatment there were on average 0.80 funding agencies other than the NIH or NSF per person for near winners; for near misses, this number is 0.81 (t -test p -value = 0.92). Between year 6 to year 10, there were on average 2.03 other funding agencies per person for near winners; for near misses, this number is 1.50 (t -test p -value = 0.40).

Approach C: manually checking acknowledge statements included in publications:

To further make sure our results are not affected by the PubMed coverage, we manually checked the acknowledgement statement of 100 random selected papers within 5 years after the treatment (50 for each group). We downloaded the PDFs, and read through them one by one. We arrived at the same conclusions (0.47 other grants per paper for near winners, 0.35 other grants per paper for near misses, p -value = 0.45).

In conclusion, our analyses provide empirical evidence that near misses were unlikely to receive more funding from agencies other than NIH or NSF.

Thank you for the insightful comment that leads us to further strengthen the conclusion. We have now added these new analyses/discussions to the paper/SI Appendix 3.6. The new text in the main text is as follows:

Although the NIH is the world's largest funder for biomedical research, near misses might have obtained more funding elsewhere. To test this hypothesis, we further collected individual grant histories for PIs in our sample from the Dimensions data, allowing us to calculate the total funding support from agencies worldwide beyond NIH. We first measured the total funding support from the U.S. National Science Foundation (NSF) received by individuals with the same name in the same period, finding near winners obtained significantly more NSF funding within 5 years after treatment (SI Appendix S3.6). We further calculated the total funding support from agencies other than the NIH or NSF, finding that near misses did not acquire more funding than near winners (SI Appendix S3.6). We also manually checked acknowledgement statements within a fraction of papers published by the two groups (SI Appendix S3.6), finding again the same conclusion.

1.3 How do the authors account for publication lags? Could it be that many of the papers published after the near miss were written before the near-miss but were moving through the publication process? One could also see where the hit papers occurred in the ten period. If they are occurring shortly after the near miss, then it may be more important to look into this. One could look at preprints or "date of acceptance" versus print.

Response: Thank you very much for this comment and suggestion. Here we build on this idea and take it further to interrogate our data, asking whether hit papers by near misses tend to appear relatively early in the post period. We consider two, related kinds of analysis.

In the first analysis, we compared the hit probability for near misses and near winners for year 0 and year 1. Given that the median time of a manuscript from submission to acceptance in a journal is around half a year⁴ and not all papers are accepted in the first submission⁵, papers published within one year after treatment may constitute those in the publication process, and are likely to have been produced before the treatment. We find that, for papers produced within one year after treatment, there is a lack of difference in the hit paper probability between near misses and near winners (14.9% for near misses versus 15.3% for near winners, χ^2 test, p -value = 0.50). This result is consistent with our RD design, indicating that the two groups should be similar across various characteristics *ex ante*, including the characteristics of works in the publication process. It also

appears to reject the hypothesis that the observed difference between the two groups is simply due to hit papers moving through the pipeline.

Building on this analysis, we next follow your suggestion and investigate where hit papers occurred within the subsequent 10-year period. We divided the 10-year window into two non-overlapping periods; specifically, from 1 to 3 years and from 4 to 10 years. If the pipeline hypothesis is correct, then hit papers by the near misses should be more concentrated in the initial period. However, when we break our data into these two periods, we find that hit papers by near misses were not concentrated in the initial period following the treatment (Fig. R1). Rather, the effect tends to be much more pronounced from year 4 to 10.

Figure R1 Comparing future career outcome between near misses (orange) and near winners (blue). (A) Near misses outperformed near winners in terms of the probability of producing hit papers in the next 1-3 years, and 4-10 years. (B) The average citations within 5 years of publication. Here we use data from 1990 to 2000 in order to eliminate boundary effects. (C) The average number of publications per person. *** $p < .001$, ** $p < .05$, * $p < .1$; Error bars represent the standard error of the mean.

Together these results show that the performance advantage by the near misses did not occur overnight, suggesting that publication lags are unlikely to explain why near misses outperformed near winners.

We also liked your idea of looking at preprints or date of acceptance data, which prompted us to dig deeper into our databases to gather this information. However, the data available to us so far, while quite extensive, were not useful in this regard: The state-of-art publication datasets don't have "date of acceptance" information. And, for preprints, *bioRxiv* didn't exist until 2013, and *ArXiv* mainly covers physics papers with a rather poor coverage for biomedical science. Hence as much as we'd like to, unfortunately we couldn't look into the preprints or date of acceptance directly, but we hope the two analyses presented above are helpful.

Thanks again for the above comment and suggestion. We now added these analyses as a separate section in SI (SI Appendix S3.7, Fig. S16 in the revised SI), and mentioned these results in the main text as well. This new text is as follows:

We also checked whether the results may be affected by pre-existing papers moving through the publication process (SI Appendix S3.7).

1.4 When setting up the RD design, were there disproportionately more papers at an early/late time period for the near-miss/near-win groups? Later time period papers likely receive more citations because there are more papers that are citing. One could set up the RD design to account for this, but it wasn't clear if the authors did this and accounted for the time period effect.

Response: Thank you very much for this comment. In the original version, we adjusted hit paper indicator/citation measures based on publication year but did not test the timing point directly. Following your suggestion, we directly compared papers' publication time by the near-miss and near-win groups. We find no substantive or statistical difference between the two groups, showing that near misses did not publish disproportionately more or fewer papers at later time periods than near winners did (t -test p -value = 0.47), which helps to alleviate the potential concern.

This result is consistent with several other results reported in the paper, which could also help address this concern. More specifically:

1. When calculating if a paper is a highly cited paper (i.e., hit paper), we control for both its *publication year* and field. This means, a paper can only be considered as a hit paper if it stands out among its contemporary peers (e.g., top 1%, top 5%, top 10%, or top 15% among all papers from the same field and the same publication year.) This method has been used widely in prior studies in the field⁶⁻⁸.
2. When calculating the normalized citations, we have also controlled its *publication year* (Fig. S12 a, c). We followed a canonical method⁹ to measure the normalized citation for each paper, by dividing the total number of citations by the average number of citations received by papers from the same field and the same year.

Finally, to ensure publication year does not affect our conclusions, we took your suggestion further and performed additional regression analysis by adding publication year fixed effects as an additional control. Publication year fixed effect allows us to estimate the effect of near miss on career impact only sampling articles published within the same year. We uncovered the same results, and find our conclusions to be robust (as shown in Fig. R2).

Thank you again for this important comment. We have now changed the Method Section in the main text, and added the Fig. R2 to the SI as a new figure (SI Appendix S3.10, Fig. S31 in the revised SI). The new text now reads:

Finally, in order to test the robustness of the regression results, we further controlled other covariates including publication year, PI gender, PI race, institution reputation as measured by the number of successful R01 awards in the same period, and PIs' prior NIH experience. We recovered the same results (Fig. S17).

Figure R2 The results from the fuzzy RDD estimation by adding publication time as fixed effects. (A) The effect of near miss on the probability to publish top 5% hit papers for applicants in the NIH system; (B) the effect of near miss on the average citations within 5 years after publication for applications in the NIH system; Here we use data from 1990 to 2000. (C - D) The same as A - B but for the conservative removal. Here, we use three different sample size, i.e., 5-score around the cutoff, 10-score from the cutoff, and 25-score from the cutoff. Here, we added publication time fixed effects as an additional control. Error bars represent the standard errors, and are clustered at individual PI level. *** $p < .001$, ** $p < .05$, * $p < .1$.

1.5 *How sensitive are the results to perturbations in fields? Identifying fields is difficult. They can change dramatically depending on which classifications are used. The authors have explored perturbations for other factors such as the percentage of hit papers, have the authors perturbed the field schemas for setting up the RDD.*

Response: Thank you for your comment. Following this advice, here we explored another field definition using the Medical Subject Headings.

As a small clarification, in the original manuscript, we followed canonical methods to define a scientific field by using the Web of Science subject categories assigned to each paper (In Web of Science, scientific journals are divided into different categories, from *Acoustics* to *Zoology*)^{9,10}. We also performed another field perturbation when comparing the Relative Citation Ratio (RCR)¹¹ between near misses and near winners (Fig. S12 b, d). The main purpose of the Relative Citation Ratio measure is to use co-citation networks to approximate the field of a paper.

Here, following your comment, we further collected Medical Subject Headings (MeSH terms) using the PubMed dataset, which is a hierarchically-organized terminology for indexing and cataloging biomedical information. More specifically, for each paper in the PubMed database, there are multiple MeSH terms to describe the paper's scope. As a result, for each publication, we retrieved all articles published in the same year with at least one identical MeSH term^{12,13}. We consider the focal paper as a hit paper if its citation is in the top 5% among all these retrieved

publications. This method defines fields on the paper level instead of journal (set) level, which differs substantially from the Web of Science subject categories. Performing this analysis, we arrived at the same conclusions, finding that our results are robust to this alternative field definition (Fig. R3).

Note that using MeSH terms as the basis for field classification requires us to focus on papers in the PubMed that have MeSH terms and can be linked to the Web of Science database, which drops around 30% of papers in our sample. Therefore, even though these analyses with new field definitions yielded the same conclusions, in the revised version we retained our previous definition as the main analyses and use this new result as robustness checks, as the former has better coverage than the latter.

Thank you again for this very helpful suggestion, which has further strengthened the paper, and the robustness of its conclusions. We now mention this new robustness check in the main paper and have added another section to the SI for these detailed analyses (SI Appendix S3.8, Fig. S18). The new text now reads:

We also varied our definition of fields using the Medical Subject Headings (MeSH) (SI Appendix S3.8).

Figure R3 Comparing near misses (orange) with near winners (blue) using MeSH terms to define fields. (A) Near misses outperformed near winners in terms of the probability of producing hit papers in the next 1-5 years, 6-10 years, and 1-10 years. (B) Average number of hit papers. (C) Average number of publications. *** $p < .001$, ** $p < .05$, * $p < .1$; Error bars represent the standard error of the mean.

1.6 *For the near-misses, how long does it take to receive the next funding grant? The authors provided some information on this but I wonder if this creates the need for subsetting RDD. For example, matching authors (both near-win and near-miss authors) that say receive a grant one year after the near-miss/near-win. This would align those grantees that receive their follow-up funding.*

Response: Thank you very much for the comment. Following your suggestion, we conducted several additional analyses, including looking at the time distribution for the next funding, and comparing near winners with near misses conditional on receiving funding *ex post*.

We first measure the time of the next funding, for both near misses and near winners (Fig. R4). We find that the two groups show a substantial difference in the year immediately after the

treatment, which is expected given our research design. Specifically, conditional on the fact that one group received ~\$1.3M for 5 years, and the other one didn't, we expect the near-miss group to be very active in seeking funding, especially with a not-too-bad proposal at hand which could be revised and resubmitted. By contrast, the near-win group should be very much *less incentivized* to seek funding because they have been “well fed”, and need to use the existing funding (Fig. R4A). Indeed, here we find that near misses are more likely to secure NIH grants within the first year of treatment, mainly due to the fact that they can revise and resubmit their previously rejected proposal, where near winners did not have much incentive to apply for new grants (Fig. R4A). Yet importantly, here we find that the difference can be mainly attributed to these revisions within the first year. And, the difference between the two groups became insignificant after removing all revisions *ex post* (Fig. R4B, p -value = 0.49).

Figure R4 The time of the next funded NIH grant. (A) The distribution of time of the next funded grants for near misses and near winners. (B) The same distribution after removing all *ex post* revision applications.

Following this comment, we tested if the results are similar after controlling for *ex post* funding status. Please note that controlling for *ex-post* features can upend causal identification, and thus this analysis must be interpreted with care. Specifically, focusing on individuals who secured funding shortly after the treatment can introduce selection bias, and this bias is asymmetric between the two groups. For example, PIs from the near-win group that had been well fed but nevertheless went on to secure more funding are likely to be especially ambitious or otherwise unusual researchers. Indeed, this selection bias can be validated by comparing pre-treatment features of the near-win and near-miss group that both secured a funding within two years of the treatment. We find that, *ex ante*, this group of near winners are indeed significantly better than near misses (Fig. R5A).

While overcoming this selection bias is difficult, we proceeded descriptively by comparing near misses ($N = 360$) with near winners ($N = 292$) conditional on securing a grant within 5 years after the treatment, which helps to “dilute” the initial selection bias. We find that in this case they are indeed much more similar *ex ante* (Fig. R5B), although the selection bias was not removed entirely (i.e., near winners had higher citations, higher probability to publish hit papers *ex ante* compared with near misses). Nevertheless, we find that *ex post*, these near misses again outperformed these near winners even when the latter were better than the former, showing a strong support for the robustness of our results against this variation (Fig. R6).

We added this robustness check to the revised manuscript and put the results in the SI (SI Appendix S3.9, Fig. S19). The new text now reads:

We further repeated our analyses by controlling *ex post* funding status for near winners and near misses (SI Appendix S3.9).

Figure R5 Comparing near misses and near winners who got a NIH funding after treatment *ex ante*. We compared various demographic and performance characteristics. The features are defined as follows (from top to bottom): 1) percentage of female applicants; 2) number of years since the first R01 application; 3) number of years since the first publication; 4) number of previous R01 applications; 5) number of publications prior to treatment; 6) number of prior papers that landed within the top 5% of citations within the same field and year; 7) probability of publishing a hit paper; 8) average citations papers received within 5 years of publication; and 9) citations normalized by field and time. Error bar represents the 95% confidence interval. (A) PIs who got a NIH funding within 2 years after treatment; (B) PIs who got a NIH funding within 5 years after treatment. *** $p < .001$, ** $p < .05$, * $p < .1$.

Figure R6 Comparing near misses (orange) with near winners (blue) conditional on securing another successful grant within 5 years of treatment. (A) Near misses outperformed near winners in terms of the probability of

producing hit papers in the next 1-5 years, and 6-10 years. (B) The average number citations within 5 years of publication. (C) The average number of publications per person. *** $p < .001$, ** $p < .05$, * $p < .1$; Error bars represent the standard error of the mean.

1.7 Grant scores are a blunt instrument for accessing why a grant was denied. Do the authors have any other information for those scores? I see so many issues that are hidden under the scores. For example, it could be that a grant was denied, not for quality issues, but for scoping issues. That gets lost in this data. It is also where I think real mechanisms reside for explaining the near misses.

Response: Thank you for this comment. NIH reviewers consider a variety of factors when judging grant proposals, including innovation, investigators, significance, approach, and environment¹⁴. Together, the judgements of these factors are combined and encapsulated into the scores. Hence you are right that, if we could obtain data beyond the scores, it would help us understand better the underlying processes. Prompted by this comment, we reached out to our NIH colleagues. It turns out, however, that there is no additional information available beyond these scores. The peer-review process at NIH is confidential, and all materials used during the peer-review process are destroyed at the end of the study section meeting, including raw scores¹⁵. In other words, even people within NIH don't have any more information than what we have, and there is not source of more or better data than what has already been offered to us. Indeed, our NIH colleagues emphasized to us that having access to scores by itself is highly unusual, and we may in fact be the only ones who currently have access to such information, which further highlights the novelty of our study.

Although it is not possible to obtain more granular data beyond the scores, we do wish to point out that there are two analyses in our paper that can speak to this issue, both of which suggest that the observed effect is unlikely to be affected by the peculiarities pertaining to the scores.

First, the design of the NIH score system indicates that NIH program directors should follow the scores recommended by reviewers, i.e., they shouldn't play favorites, especially for those on the margin^{16,17}. This can also be validated using pre-treatment features to test if there are observable differences between the two groups. In this paper, we tested 11 different dimensions that cover a broad range of characteristics for these applicants, and found that across all these dimensions, they are indeed statistically indistinguishable (Fig. 1b in the main text). Related, when we present this paper to NIH colleagues, they recognize this finding as reflecting their explicit policies against bias.

One limitation of the approach above is that we can only test characteristics that are observable. But it could be that what separated winners from losers were not observable. This is where our second analyses based on regression discontinuity design can help. Indeed, as discussed in detail in Method **Econometric model specification and estimation procedures**, the regression discontinuity design helps us account for all factors that differ smoothly with the score, ensuring that applications just above and just below the payline (as opposed to funded or not) don't differ

in any observable and unobservable ways. The agreement between the RD approach and that directly compares near misses and near winners, further confirms the consistency of our results.

1.8 Small thing: The “what doesn’t kill you, makes you stronger” quotes are a little overdone. I think the point is made strongly enough. You could probably remove one or more of the quotes.

Response: Thank you, and we agree. In the revised manuscript, we have removed the quote in the Discussion section, while retaining the one in the introduction as part of the existing literature.

Thank you again for all your helpful comments and suggestions. They have inspired many new analyses that we believe have led to a stronger paper. Please do not hesitate to let us know if there is anything further we can do to improve this work.

Referee #2:

2.1 *The authors empirically compared two groups of applicants applying for U.S. National Institutes of Health (NIH) R01 grants: “near-miss” and “near-win”. The study is interesting and methodologically sound. However, it seems that the authors have not searched previous peer review literature systematically. Studies with similar results (similar ex-post performance of best-rejected and approved applicants) have been published earlier (e.g., Melin & Danell, 2006). An overview of studies can be found in Bornmann (2011). The following points should be considered in a revision of the manuscript:*

Response: Thank you very much for the positive comments about the interest and methodological soundness of the paper. The pointers to existing work are very helpful, and thank you for bringing these references to our attention.

In the revised version of the manuscript, we have added more theoretical backgrounds and cited relevant papers from the peer review literature as guided by Bornmann (2011). We have also added other appropriate references to prior literature in the main text.

One side note is that the nice paper by Melin & Danell¹⁸ showed that securing past grants is associated with more future funding, which is consistent with the grant dynamics we see in our data. We now cite this paper in the introduction. But Melin & Danell also found that successful applicants published in better journals, produced more patents, and developed more spin-off research groups and firms than rejected ones, documenting that the successful group were more successful in many different ways. Hence their results are not similar to our main story but similar to the broader prior literature around the Matthew Effect and related concepts, which therefore further distinguishes the surprising nature of our results, and the contribution of our paper to this extant and canonical literature.

Paper

2.2 *Page 2: Another possible explanation for the results might be that near-miss applicants searched for signs of success (e.g., publications in high-impact journals) more intensively than near-wins, since near-wins are able to demonstrate their success by receiving the R01 grants. We possibly know this effect from statements by Nobel Laureates in which they refrain from publishing in high-impact journals. They can say that: since they received the Nobel Prize, it is not necessary for their career to focus on signs of academic success besides the prize.*

Response: Thank you for this helpful comment, which raises an interesting hypothesis: near winners have demonstrated their success by receiving the R01 grants, whereas near misses searched for signs of success more intensively. Although we can't test this hypothesis directly in observational data like ours, it is indeed an interesting hypothesis, and we fully agree that such hypotheses should be considered in the paper as well.

Indeed, the goal of this paper is to establish the empirical fact that early-career setbacks are linked to longer-term career impacts. We are delighted that all three referees have appreciated this

contribution. But at the same time, the set of underlying mechanisms are difficult to decode, as the effect is likely due to a combination of multiple mechanisms and we can only test mechanisms that are observable (Fig. S28). This hypothesis for example adds to other plausible hypotheses that are unobservable as well, and we fully agree that it should be discussed properly in the paper.

We have now added this interesting hypothesis to the discussion section of the revised manuscript, and feel that as a result, the paper has further improved in its soundness and also helps direct future work in this area. The new text now reads:

More generally, these numerous observable features considered do not account alone or collectively for the performance change, suggesting that unobservable dimensions may play a role, including effort, signaling, and grit factors following setbacks or sacred sparks.

2.3 Page 2: One should consider in the interpretation of the results that the authors analyzed a highly selective group of scientists. Setbacks might be goosing only for people who know their potential (near-miss applicants, because they already had success otherwise), but might have different effects on other scientists (e.g., disillusioning or frustrating).

Response: We fully agree with this comment—the observed effect is for a specific group of NIH PIs, i.e., PIs that are good enough to be around the payline. As with any RD analyses (as with natural and field experiments), the effect mainly pertains to the specific population on which the experiment was conducted. We do wish to add that we arrived at the same conclusion when we focus on PIs further from the payline by adding proposal score as a control variable in our RD analyses (Method section in the main text, and Fig. S7 in the Supplementary Information), which to a certain degree helps to alleviate this concern. Even with this generalization, we cannot say for sure if this effect holds for other PIs, such as those with very good or bad scores (since we do not have appropriate counterfactuals). But we agree that this is an important point that needs to be emphasized.

In the revised manuscript, we have added specific discussions in the paper to highlight the fact that we mainly focused on the population around the payline, which may or may not generalize to other populations. We feel that, thanks to your comment, these discussions further tightened the presentation of our results. The new text now reads:

As with any RD analysis, the effect pertains to the specific population on which the experiment was conducted. While the RD approach allows us to expand our sample to a wider range of setbacks by controlling for evaluation scores, yielding the same conclusions, to what degree our findings may generalize substantially beyond near misses is a question we cannot yet answer conclusively.

2.4 Page 3: There are three theories available to explain success or failure in scientific careers: the “sacred spark” theory (Cole & Cole, 1973), the “accumulative advantage” theory (Merton, 1968), and the “utility maximizing theory.” According to the last theory, “all researchers choose to reduce their research efforts over time because they think other tasks may be more advantageous” (Kwiek, 2015). All these theories should be considered in the manuscript under review.

Response: Thank you for this excellent suggestion. We now introduce all these theories in the revised paper, and cited relevant literature.

2.5 Page 3: *The authors have an interesting study design which can be improved, however, in my opinion: (1) the authors included mostly performance indicators in the comparison of near-miss and near-win groups and only a few demographics. I recommend to include a broader set of variables considered in the comparison (nationality, reputation of previous institutions, number of children etc.). Factors should be considered, which do not only have a possible effect on peer review decisions (thus, take a look at the corresponding bias literature, see Bornmann, 2011), but also on careers of scientists. (2) The design is not convincing: The authors should ensure that both groups (near-miss and near-win) are really similar. In order to investigate in a quasi-experimental evaluation study the outcome of “social intervention programs”, Rossi, Freeman, and Lipsey (1999) favor the “comparison group design”: “Targets receiving the program are compared with some selected group of targets or potential targets who do not receive the program” (p. 310). They recommend to generate pairs of two people (so-called “artificial twins”) for a treatment group (people receiving the program) and a control group (people not receiving the program). Both groups should be as similar as possible. For example, Baker, Robertson, and Toguchi (1996) selected for APRA fellowship holders “a matched sample of non-APRA holders” (p. 24).*

Response: Thank you for this very useful comment. After carefully looking into the peer review literature suggested, we find that, in addition to what has been controlled in the original manuscript, we were able to include other potentially relevant factors^{19,20}, including the reputation of the scientific institution to which an applicant belongs, and applicant’s gender, nationality/ethnicity. We also notice that some of the demographic data such as the number of children are difficult to collect, as also pointed out by the Editor. Therefore, to follow your advice and that of the Editor, and to further control additional features, here we considered applicant’s gender, institutional reputation (as measured by the number of R01 grants awarded between 1990 to 2005) and PI ethnicity as additional features. The reason we focused on ethnicity instead of nationality is because our data mainly contain U.S. based scientists. We used a start-of-art ethnicity detector based on the first and last name²¹, and applied it to the PIs in our sample. By computing these features for the PIs in our study, we find that there is no significant difference between near misses and near winners in terms of either institutional reputation (p -value = 0.68) or ethnicity (p -value = 0.15) or gender (p -value = 0.14, Fig. 1b) *ex ante*.

We have now added this comparison to the main figure of the revised manuscript (Fig. 1b).

To ensure our results are not affected by these additional demographic features, we further controlled these features (i.e., gender, ethnicity, and institutional reputation) in the 2SLS regression model, finding our results to be robust (Fig. R7). We have also added this result to the SI (SI Appendix S3.10, Fig. S17).

Figure R7 The result from the fuzzy RDD estimation after controlling demographic features and publication years. (a) The effect of near miss on the probability to publish top 5% hit papers for applicants in the NIH system; (b) the effect of near miss on the average citations within 5 years after publication for applications in the NIH system; Here we use data from 1990 to 2000. (c) The effect of near miss on the number of hit papers for applicants in the NIH system. (d - f) The same as a - c but for the conservative removal. Here, we use three different sample size, i.e., 5-score around the cutoff, 10-score from the cutoff, and 25-score from the cutoff. Here, we added demographic feature and publication time fixed effects in addition to all the controls we have in the main text. Error bars represent the standard errors, and are clustered at individual PI level. *** $p < .001$, ** $p < .05$, * $p < .1$.

Another important comment and suggestion is to use a matching strategy to ensure the similarity between near misses and near winners. Inspired by this comment, in the revision we carried out a new set of experiments by using the state-of-art matching method, i.e., “Coarsened Exact Matching (CEM)²²”, which helps to ensure that near misses and near winners are observably similar *ex ante*. The CEM algorithm involves three steps²³:

1. Temporarily coarsen each control variable X , for the purposes of matching;
2. Sort all observations into strata, each of which has the same values of the coarsened X .
3. Prune from the dataset the units in any stratum that do not include at least one treated and one control unit.

Following the algorithm, we use a set of *ex ante* features to control for individual grant experiences, scientific achievements, demographic features, and academic environments; these features include the number of prior R01 applications, number of hit papers published within three years prior to treatment, PI gender, ethnicity, reputation of the applicants’ institution. In total, we matched 475 of near misses out of 623; and among all 561 near winners, we matched 453. The overall balance was improved (L_1 statistics²⁴ reduced from 0.61 to 0.45), suggesting the validity of the method. We first show that the difference between matched near misses and near winners are indeed similar in terms of all observable dimensions (Fig. R8), with all measurements being statistical indistinguishable (p -value > 0.1). We then repeated all our analyses, finding within the matched

samples, near misses again outperformed near winners in the subsequent 10 years after treatment, and the difference cannot be fully explained by the screening effect (see Fig. R9).

We wish to thank you for this great suggestion. We feel that this new analysis using a matching procedure further strengthened the conclusions of the paper. Indeed, with the addition of this new analysis, we now present three different approaches to document the same effect, each with their own strengths (the direct comparison approach provides analysis of raw data and is intuitive; the matching approach developed thanks to your comment controls across observable dimensions; and the RD approach further allows us to draw stronger causal conclusions by eliminating the potential influence of both observable and unobservable factors.) Together the consistency across all three approaches helps support the robustness of our findings.

We now mention explicitly this new analysis in the paper, and describe the matching strategy in the Method section in detail. We have also added a new section in the SI (SI Appendix S3.10, Fig. S20 and Fig. S21) to describe the results.

Figure R8 Comparing near misses (orange) and near winners (blue) *ex ante* using the CEM matching. Pre-treatment feature comparisons between the near-miss and near-win group. We compared various demographic and performance characteristics. The features are defined as follows (from top to bottom): 1) percentage of female applicants; 2) number of years since the first R01 application; 3) number of years since the first publication; 4) number of previous R01 applications; 5) number of publications prior to treatment; 6) number of prior papers that landed within the top 5% of citations within the same field and year; 7) probability of publishing a hit paper; 8) average

citations papers received within 5 years of publication; and 9) citations normalized by field and time. We see no significant difference between the two groups across any of the ten dimensions we measured; Error bar represents the 95% confidence interval.

Figure R9 Comparing near misses (orange) with near winners (blue) using CEM matching. (A) Near misses outperformed near winners in terms of the probability of producing hit papers. (B) The average number citations within 5 years of publication. Here we use data from 1990 to 2000. (C) The average number of publications per person. (D - F) The same as A - C but for the conservative removal. *** $p < .001$, ** $p < .05$, * $p < .1$; Error bars represent the standard error of the mean.

2.6 Page 6: *The authors write: “All else equal, except that one has had an early funding failure and the other an early funding success, it is the one who failed that is expected to write higher-impact papers in the future.” In my opinion, this conclusion is too strongly formulated. The probability is higher, but highly cited papers can be expected from near-win applicants too (with higher probability than expected).*

Response: Thank you for pointing this out. We have now edited this passage in the revised manuscript, mentioning this point specifically and aiming for greater clarity. The passage now reads:

This finding itself has a striking implication. Indeed, take two researchers who are seeking to continue their careers in science. While both near-miss and near-win applicants published high-impact papers at a higher rate than their contemporary peers, comparing between the two groups, it is the ones who failed that are more likely to write a high-impact paper in the future.

2.7 Page 9: *The authors write: “they represent an imperfect proxy at best to measure scientific outputs, suggesting that future research may fruitfully explore measurements that extend beyond*

citation-based measures to examine broader career outcomes”. In my opinion, this should not only be a point in the discussion section, but should be done by the authors in their study. Please use other indicators different from bibliometrics to validate your results.

Response: Thank you for this very insightful comment. Here you have asked an important question: Can we use other indicators beyond bibliometrics to further compare the performance of near misses and near winners, especially in dimensions that go beyond citation metrics? Indeed, if we could examine broader outcomes, it would really push the frontier. It can also get to the heart of the NIH’s mission, which is to “*seek not only the fundamental knowledge about the nature and behavior of living systems but also the application of that knowledge to enhance human health*” (<https://www.nih.gov/about-nih/what-we-do/mission-goals>).

In this revision we collected new datasets that allowed us to go significantly beyond citation metrics by developing three different indicators, specifically focusing on indicators that are aligned with NIH missions (i.e., bench to bedside translations):

1. The first indicator is whether a paper is a clinical trial publication as indicated in the PubMed database, representing direct contribution to clinical trials;
2. The second indicator is whether a paper has been cited by clinical trial publications, representing indirect contribution to clinical trials;
3. And the third one is a new identifier developed by our NIH colleagues to quantify whether a paper has the potential to be translational research. Specifically, the Approximate Potential to Translate (APT) score was used to identify early signatures of bench-to-bedside translation^{25, 26}. The score is estimated using machine learning method combining features such as MeSH terms, disease, therapies, chemical/drug, and citation rates.

We compared near misses with near winners in the subsequent ten-year window across these three new dimensions. We find that near misses systematically outperformed near winners across all dimensions (Fig. R10). Finally, to test if the tendency toward clinical research can by itself account for the observed citation difference between near misses and near winners, we separated their publications into clinical and non-clinical papers, finding that within non-clinical papers, near misses again outperformed near winners (Fig. R10 D, H). We are very excited about these new results, and feel that they have made our paper stronger, thanks to your insightful suggestions. To this end, we decided to add the new results to the main paper as a new figure by itself, with associated discussions regarding how the works done by the near misses have clinical relevance that went beyond citation metrics. The new text reads:

While citations and their variants have been used extensively to quantify career outcomes, they may represent an imperfect or limited proxy for measuring output, prompting us to ask if the observed effect of early-career setback extends beyond citation measures. To this end, we used additional datasets to calculate three indicators probing the clinical relevance of the works. These measures are: 1) whether a paper is a clinical trial publication (direct contribution to clinical translation); 2) whether a paper has been cited by at least one clinical trial publication (indirect contribution to clinical translation), and 3) whether a paper has potential to become translational research (potential for translation). We compared works produced by near misses and near winners over the ten-year period,

finding that across all three translational dimensions, near misses systematically outperformed near winners. Specifically, near misses were 50% more likely to publish a clinical trial paper compared with near winners (p-value < 0.001, Fig. 4a), and their overall publications were 19.6% more likely to be cited by clinical trials (p-value < 0.001, Fig. 4b), and are 24.5% higher in their potential for bench-to-bedside translation (p-value < 0.001, Fig. 4c). We also find that, all these conclusions remain the same after conducting the conservative removal procedure as described in Fig. 3 (p-value < 0.001, Fig. 4e, f, g). Finally, to test if the tendency toward clinical research can by itself account for the observed citation difference between near misses and near winners, we separated their publications into clinical and non-clinical papers, finding that within non-clinical papers, near misses again outperformed near winners (Fig. 4d, h). We repeated all our analyses using the RD approach, recovering broadly consistent conclusions (SI Appendix S3.10). Together, the results shown in Fig. 4 suggest that the uncovered effect of early-career setback goes beyond citation measures, with near misses outperforming near winners in both basic and translational science.

Figure R10 Near misses (orange) outperformed near winners (blue) in both basic and translational science. Here we considered three measures probing the clinical relevance of their research: 1) clinical trial papers in the PubMed dataset (direct contribution to clinical translation); 2) papers cited by at least one clinical trial paper (indirect contribution to clinical translation); 3) papers with potential to become translational research. Specifically, the approximate potential to translate (APT) score was used to identify early signatures of bench-to-bedside translation. The score is estimated using machine learning combining features such as MeSH terms, disease, therapies, chemical/drug, and citation rates⁷⁷. (a) Near misses outperformed near winners in terms of the probability of producing clinical trial papers in the next 1-5 years, and 6-10 years; (b) The same as a but for papers cited by clinical trials in the future; (c) The same as a but for papers with potential to be a translational research; (d) Hit paper probability by considering non-clinical trial papers only. (e - h) The same as a - d but for the conservative removal (following the same procedure in Fig. 3b). *** p < .001, ** p < .05, * p < .1; Error bars represent the standard error of the mean.

2.8 Figure 1: The case numbers are mentioned below the figure (561 near-win and 623 near-miss applicants), but not in the text. However, this is an important information.

Response: Thank you, and we fully agree that we should have made this fact clearer in the paper. In the revised main text, we emphasized specifically this point by discussing these numbers directly in the main text. Now the new text reads:

In total, our analyses yielded 561 near winners and 623 near misses around the payline.

Supporting Information

2.9 Page 3: *It is no longer Thomson Reuters, but Clarivate Analytics.*

Response: Thank you for pointing this out. We have corrected this statement in the revised main text and SI.

2.10 Page 4: *How are fields defined? The best field categorization scheme which can be used for comparison of citation impact might be MeSH terms (Bornmann, Marx, & Barth, 2013). Bibliometric studies have pointed out several weaknesses of using journal sets as field categorization scheme.*

Response: Thank you for the comment. In the original manuscript, we followed canonical methods to define a scientific field using the Web of Science subject categories assigned to each paper (In Web of Science, scientific journals are divided into different categories, from *Acoustics* to *Zoology*)^{9,10}. Within each category a list of journals is provided. And, hit papers were defined as being in the top citations received in the same *field* and *time*, following existing studies^{9,10}.

Here, in order to further make sure our results are not affected by the definition of field, we followed your comment, and collected additional data on MeSH terms, which is a hierarchically-organized terminology for indexing and cataloging biomedical information. The goal here is to use the MeSH terms to define fields and then repeat our analyses to see if our results still hold. More specifically, for each paper in the PubMed database, there are multiple MeSH terms to describe the paper's scope. As a result, for each publication, we retrieved all articles published in the same year with at least one identical MeSH term¹². We consider a paper as a hit paper if it is in the top 5% of citations received among all these retrieved publications. This method defines fields on the paper level instead of journal (set) level, which differs substantially from the Web of Science subject categories. Performing this analysis, we arrived at the same conclusions, finding that our results are robust to this alternative field definition (Fig. R11).

Note that using MeSH terms as the basis for field classification requires us to focus on papers in the PubMed that have MeSH terms and can be linked to WOS, which drops around 30% of papers in our sample. Therefore, even though these analyses with new field definitions yielded the same conclusions, in the revised version we retained our previous definition as the main analyses and use this new result as robustness check, as the former has better coverage than the latter.

Figure R11 Comparing near misses with near winners using MeSH terms to define fields. (A) Near misses outperformed near winners in terms of the probability of producing hit papers in the next 1-5 years, 6-10 years, and 1-10 years. (B) Average number of hit papers. The near-miss applicants again outperformed their near-winning counterparts. (C) Average number of publications. *** $p < .001$, ** $p < .05$, * $p < .1$; Error bars represent the standard error of the mean.

Thank you again for this suggestion, which has further strengthened the paper, and the robustness of its conclusions. We now mention this new robustness check in the main paper, and added a separate section to the SI (SI Appendix S3.8).

2.11 Pages 5 and 6: *Tekles and Bornmann (2019) investigated several methods for author disambiguation. Their analyses show that the method published by Caron and van Eck (2014) performs best. The authors of the manuscript under review should orient towards such analyses to decide which disambiguation method is used in their study. Furthermore, a sample of PIs should be drawn and manually checked whether the disambiguation method works satisfactorily or not.*

Response: Thank you for this valuable suggestion. Following your suggestion, we implemented the method by Caron and van Eck^{27,28}. This nice method uses unsupervised learning to cluster potential papers using information including common references, common citations, common addresses, email, journal and subject categories, etc²⁸. Two papers are clustered if the score exceeds a certain threshold. After clustering, we took a step further to compare papers from each group with NIH associated publications by the considered PI (these publications represent ground-truth information that none of the existing disambiguation algorithms had access to). If a PI has at least one NIH-related publication, all groups that contain these publications are associated with this PI. We conducted the same analysis as the main text, finding our results robust (Fig. R12).

Further, to ensure the disambiguation method in the main text works well, we randomly select 1,000 author-publication pairs from PIs around the payline. Specifically, we randomly select 50 PIs, and 20 publications for each PI. By removing meeting abstracts in the Web of Science data, we obtained 576 positive cases in which papers were published by the PI, and 225 negative cases in which papers were written by different persons with similar name as the PI. We then performed manual search for PIs' websites, Google Scholar profiles if any, affiliations, coauthors etc. Out of these 801 pairs, we find the false positive rate (i.e. fraction of times the algorithm indicates the paper belonging to the PI, while they do not) to be 8.9% and a false negative rate (i.e. fraction of times that the algorithm considers the paper belonging to a different person, while they do not) of 4.7%. These low error rates support the validity of the disambiguation algorithm we used.

Although name disambiguation methods are becoming increasingly accurate, the main problem is that we often don't know how perfect or imperfect they are. Yet, while the name disambiguation methods may be imperfect, it's important to note that there are also reasons to believe that potential errors in name disambiguation methods do not affect our conclusions, as such errors are likely to differ smoothly with the score hence are controlled for by our RD analyses. In other words, the errors don't favor any particular group above or below the payline. Nevertheless, our goal here is to try several state-of-art name disambiguation methods to further ensure the robustness of our results (Fig. S2 and Fig. S3).

We added the results from the new name disambiguation method suggested by the referee in the SI (SI Appendix S1.4, Fig. S2) and mentioned it in the main text.

Figure R12 Comparing future career outcome between near misses (orange) and near winners (blue) using the name-disambiguation method proposed by Caron and van Eck²⁸. (A) Near misses outperformed near winners in terms of the probability of producing hit papers in the next 1-5 years, 6-10 years, and 1-10 years. (B) Average citations within 5 years of publication. The near-miss applicants again outperformed their near-winning counterparts. To ensure all papers have at least 5 years to collect citations, here we used data from 1990 to 2000 to avoid any boundary effect. (C) The average number of publications per person. (D - F) The same as A – C but for the conservative removal. *** $p < .001$, ** $p < .05$, * $p < .1$; Error bars represent the standard error of the mean.

2.12 Pages 11 and 12: It is not clear how fields have been defined.

Response: Thank you for bringing this issue to our attention. For clarification, here we followed canonical studies, and used Web of Science subject categories to define fields (In Web of Science, scientific journals are divided into different categories, from *Acoustics* to *Zoology*)^{9,10}.

We now clarified how fields are defined in the revised main text and SI.

2.13 Page 12: *The authors used the RCR indicator in their study: The indicator has been critically assessed in bibliometrics (Waltman, 2015).*

Response: Thank you very much for pointing out this issue. We completely agree—while the RCR measure developed by our NIH colleagues represents an interesting attempt to normalize citations across different fields, the research community is still working through its various implications and has yet to reach a firm conclusion. Here our goal is not to take a stand on whether the RCR is *the* measure for controlling for field variations, but rather use it as *a* measure to further stress test our results. While the RCR measure -- like any bibliometric measure -- may have its own limitations as discussed in Waltman 2015, here it is reassuring to know that our results still hold using the RCR index, further showing the robustness of our results. At the same time, it's important to make sure that our results do not depend on the RCR index specifically, which is why we have also tried other measures such as C_5 (the number of citations within 5 years of publication), c_f (the normalized citations by field and time⁹). The additional new metrics using clinical-trial applications (see response above) provide further and distinct alternatives.

We have now cited Waltman2015 when introducing the RCR measurement to represent a more balanced view of the literature.

2.14 Reference:

1. Baker, M., Robertson, F., & Toguchi, H. (1996). *The Australian postgraduate research award scheme: An evaluation of the 1990 cohort*. Canberra, Australia: Higher Education Division.
2. Bornmann, L. (2011). *Scientific peer review*. *Annual Review of Information Science and Technology*, 45, 199-245.
3. Bornmann, L., Marx, W., & Barth, A. (2013). *The normalization of citation counts based on classification systems*. *Publications*, 1(2), 78-86.
4. Caron, E., & van Eck, N. J. (2014). *Large scale author name disambiguation using rule-based scoring and clustering*. In P. Wouters (Ed.), *Proceedings of the science and technology indicators conference 2014 Leiden "context counts: Pathways to master big and little data"* (pp. 79-86). Leiden, the Netherlands: University of Leiden.
5. Cole, J. R., & Cole, S. (1973). *Social stratification in science*. Chicago, MA, USA: The University of Chicago Press.
6. Kwiek, M. (2015). *The european research elite: A cross-national study of highly productive academics in 11 countries*. *Higher Education*, 71(3), 379–397.
7. Melin, G., & Danell, R. (2006). *The top eight percent: Development of approved and rejected applicants for a prestigious grant in sweden*. *Science and Public Policy*, 33(10), 702-712.
8. Merton, R. K. (1968). *Social theory and social structure*. New York, NY, USA: Free Press.
9. Rossi, P. H., Freeman, H. E., & Lipsey, M. W. (1999). *Evaluation: A systematic approach*. London, UK: Sage Publications.
10. Tekles, A., & Bornmann, L. (2019). *Author name disambiguation of bibliometric data: A comparison of several unsupervised approaches*. <https://arxiv.org/abs/1904.12746>
11. Waltman, L. (2015). *NIH's new citation metric: A step forward in quantifying scientific impact?* Retrieved November 10, 2015, from <http://www.cwts.nl/blog?article=n-q2u294&title=nihs-new-citation-metric-a-step-forward-in-quantifying-scientific-impact#sthash.w1KC3A1O.dpuf>

Response: Thank you very much for pointing out these very helpful references. We have cited all relevant references you mentioned (including Bornmann2011, Bornmann2013, Cole1973, Caron2014, Kwiek2015, Melin2006, Tekles2019, and Waltman2015).

Thank you again for all your helpful and insightful comments and suggestions. They have inspired many new analyses that we believe have led to a stronger paper. Please do not hesitate to let us know if there is anything further we can do to improve this work.

Referee #3

This paper studies the effect of early career setbacks (i.e. failure in NIH R01 grant applications) on career development and finds that “near misses” outperform “near wins,” suggesting “what doesn’t kill me makes me stronger.” The finding is very novel and has important policy implications. The research is rigorously designed and executed, carefully ruling out various alternative explanations and implementing various robustness tests. The paper is definitely an important contribution and should be published.

Response: Thank you very much for these positive comments about our paper. Please see below for the further analyses we have undertaken in response to your insightful comments.

On the other hand, I also have some comments and suggestions.

3.1 *Comparing “near miss” and “near win” and regression discontinuity design allows the authors to make strong causal claims. However, one important limitation is that it is only about the local effect around the cut-off point. In other words, the findings might not be generalizable to applicants who have a very high review score, namely we do not know whether declining their funding will actually help their performance. Furthermore, in this special context, we might expect that the local effect is very special. Specifically, there might be an alternative explanation: it might not be about setbacks but rather about “undeserved success and luck.” The grant review process is supposed to screen out “unworthy” projects. Near misses and near wins are both “sub-optimal” projects. Failing the grant application allows the candidate to further develop their ideas or change to more promising ideas, while successful applicants commit to sub-optimal ideas and miss the opportunity of finding better directions. Therefore, it is actually about “underserved success” rather than setbacks. The robustness tests (S3.10) is related to this, but it is in line with this alternative explanation rather than rejecting it. The authors should address this alternative explanation.*

Response: Thank you for this important comment. We agree that the observed effect is for a specific group of NIH PIs, i.e., those near the payline, and our results are thus about the local effect around the cut-off point. Indeed, as with any RD analyses (and with natural and field experiments methods), the effect can only be interpreted for the specific population on which the experiment was conducted. Therefore, even though we can control for proposal scores in our two-stage regression analysis, which allows us to investigate PIs further from the payline, we cannot say for sure if this effect holds for other PIs, such as those with very good or bad scores (since we do not have appropriate counterfactuals).

Additionally, you suggested an interesting alternative explanation, where near winners committed to their initial proposed “sub-optimal” ideas. Setbacks, on the other hand, enabled near misses to further develop their ideas or change to more promising directions. This is indeed an interesting hypothesis. We agree that our analyses in the original SI speak to this hypothesis but do not offer a more direct test.

Here we build on this idea to further interrogate our data. We have considered what kinds of analyses may help test this hypothesis and considered two predictions that may follow in line with this hypothesis:

- First, if near winners were initially committed to the “*sub-optimal*” ideas, it may suggest that their work in the more distant future would be significantly better than the research performed under this grant. We can therefore test this prediction by comparing their initial publications in the years after the grant with their later publications.
- Second, since the initial quality of these two groups are not different under the assumption of the RD (and reinforced by the tests on observables), after the grant that initially “locked in” a near winner expires, later works by near winners may be more similar to those by near misses. This suggests that under the hypothesis we may observe a convergence of career impact between these two groups beyond the initial five years.

To test these predictions, we first compare publications by the near winners within the first 5 years with those published in the next 5 years. Papers published within the first 5 years were more likely to be the product of the original proposal, whereas later papers were more likely to be related to new ideas developed beyond the initial grant. After comparing their performance across these two periods, we found an at most modest and not statistically significant improvement for near winners in terms of probability to publish hit papers (p -value = 0.14, Fig. R13A), or normalized citations (p -value = 0.13, Fig. R13B), showing little support for the first prediction.

Furthermore, we find that there is no significant difference between publications from 1 to 5 years and 6 to 10 years for near misses, indicating their performance on these dimensions are stable (and consistent with results reported in Fig. 2 of the manuscript). Importantly, there is still a significant difference between near winners and near misses between years 6 to 10, which also failed to support for the second prediction.

Together, these results (shown in Fig. R13) are consistent with those reported in the paper, but run counter to the hypothesis that near winners were temporarily “locked in” to bad ideas whereas near misses were set free. Note that, there could be path dependencies that may potentially explain the long-run differences observed in Fig. R13, but nevertheless, these analyses are also consistent with other mechanisms tested in the original manuscript. For example, we did not find any significant change in impact associated with shifting future research directions (SI Appendix S4.5). Together these results suggest that being initially trapped by the grant topic is unlikely to be sufficient to account for the performance gap.

Incidentally, it may also be worth noting that proposals near the payline do not appear to be bad ideas in an absolute sense. NIH R01 grants are highly competitive, and it’s challenging to be scored near the payline (and not just from our personal experience...) Indeed, PIs around the payline, either near-miss or near-win, are significantly stronger than scientists in the same field and year and produce papers with much higher hit rates than the baseline average (e.g., Fig. 2b). It therefore suggests that, even if the near winners were locked in by the initial grant, the initial idea does not appear to be low quality, and it may be better or worse than the next idea one may come up with.

Figure R13: Comparing future career outcome between publications within 5 years and from 6 to 10 years after the treatment, for both near misses and near winners. (A) Comparing hit rate probability, and we find there is no statistical significant difference between publications from the first 5 years and those from the second 5 years. (B) The same as A but for the average normalized citations per paper. We find there is no significant improvement for near winners. *** $p < .001$, ** $p < .05$, * $p < .1$; Error bars represent the standard error of the mean.

Thank you again for this great suggestion. We now added these analyses as a separate section in SI (SI Appendix S3.14), and mentioned these results in the main text. The new text now reads:

To test the hypothesis that near winners were committed to initially proposed ideas, we compared articles by near winners published in 5 years after treatment with those published between 6 to 10 years. We find no statistically significant improvement for near winners in terms of probability to publish hit papers (p -value > 0.1) or normalized citations (p -value > 0.1).

In addition to these new analyses, we now also emphasize more clearly in the manuscript one limitation inherent to RD designs, where the findings help isolate causation but may not generalize to populations further from the payline. Thanks to your comment, this expanded discussion has further tightened the interpretation of our results. The new text now reads:

As with any RD analysis, the effect pertains to the specific population on which the experiment was conducted. While the RD approach allows us to expand our sample to a wider range of setbacks by controlling for evaluation scores, yielding the same conclusions (see Methods section), to what degree our findings may generalize substantially beyond near misses is a question we cannot yet answer conclusively.

3.2 One complementary analysis might be: using whether having a failed grant application before (or the number of failed attempts) as an independent variable, and then examine how it affects evaluation score, and conditional on having the same score, how does it affect further career development.

Response: Thank you for this comment. We have now followed your suggestion and investigated whether having a failed grant application before affects the evaluation score. First, we find that there is a lack of difference in scores between PIs who had prior failed experience and those who did not (p -value = 0.20). Second, we go one step further to control for individual prior grant

performance. Specifically, we sorted all PIs into three different groups: PIs without prior grant applications, PIs whose most recent applications were discussed on the review panel (have scores), and PIs whose most recent applications were not discussed on the panel (no score, indicating that the application didn't pass the initial screening). By adding PI's past grant history fixed effects that only compare PIs within the same group, we find our results again to be robust (Fig. R14). Together these results suggest that the performance advantage by the near misses cannot be explained by differences in prior grant applications.

These new results are consistent with other results documented in the manuscript, which offer additional robustness checks. One important analysis in the paper is that we restricted our analyses to PIs without any prior R01 applications (first timers) representing a “cleaner” control for factors related to prior scores/grants, and we arrived at the same conclusions (Fig. S9). Moreover, when setting up the RD design, we also added polynomial controls for the current score (both linear and quadratic score controls), which helps to further control for individual differences.

In summary, we agree that adding prior grant application controls is helpful and important, and thank you for raising this comment. We have added these new analyses to the paper by including the discussion in the revised manuscript and detailed results in the SI (SI Appendix S3.10). Now the text reads:

We also controlled for fixed effects categorizing PIs' prior NIH experience, recovering the same conclusions (SI Appendix S3.10, Fig. S31).

Figure R14 The result from the fuzzy RDD estimation by controlling for fixed effects categorizing PIs' prior NIH grant experience. (A) The effect of near miss on the probability to publish top 5% hit papers for applicants in

the NIH system; (B) the effect of near miss on the average citations within 5 years after publication for applications in the NIH system; Here we use data from 1990 to 2000. (C - D) The same as A - B but for the conservative removal. Here, we use three different sample size, i.e., 5-score around the cutoff, 10-score from the cutoff, and 25-score from the cutoff. Here, we have added prior grant experience fixed effects in addition to all the controls we have in the main text. Error bars represents the standard errors, and are clustered at individual level. *** $p < .001$, ** $p < .05$, * $p < .1$.

Thank you again for your helpful comments and suggestions. They have inspired new analyses that we believe have led to a stronger paper. Please do not hesitate to let us know if there is anything further we can do to improve this work.

Reference

- 1 Way, S. F., Morgan, A. C., Clauset, A. & Larremore, D. B. The misleading narrative of the canonical faculty productivity trajectory. *P Natl Acad Sci USA* **114**, E9216-E9223 (2017).
- 2 Way, S. F., Morgan, A. C., Larremore, D. B. & Clauset, A. Productivity, prominence, and the effects of academic environment. *Proceedings of the National Academy of Sciences* **116**, 10729-10733 (2019).
- 3 Clauset, A., Arbesman, S. & Larremore, D. B. Systematic inequality and hierarchy in faculty hiring networks. *Science Advances* **1** (2015).
- 4 Powell, K. The Waiting Game. *Nature* **530**, 148-151 (2016).
- 5 Calcagno, V. *et al.* Flows of Research Manuscripts Among Scientific Journals Reveal Hidden Submission Patterns. *Science* **338**, 1065-1069 (2012).
- 6 Ahmadpoor, M. & Jones, B. F. The dual frontier: Patented inventions and prior scientific advance. *Science* **357**, 583-587 (2017).
- 7 Wang, D., Song, C. & Barabasi, A. L. Quantifying Long-Term Scientific Impact. *Science* **342**, 127-132 (2013).
- 8 Wuchty, S., Jones, B. F. & Uzzi, B. The increasing dominance of teams in production of knowledge. *Science* **316**, 1036-1039 (2007).
- 9 Radicchi, F., Fortunato, S. & Castellano, C. Universality of citation distributions: Toward an objective measure of scientific impact. *P Natl Acad Sci USA* **105**, 17268-17272 (2008).
- 10 Wang, J., Veugelers, R. & Stephan, P. Bias against novelty in science: A cautionary tale for users of bibliometric indicators. *Research Policy* **46**, 1416-1436 (2017).
- 11 Hutchins, B. I., Yuan, X., Anderson, J. M. & Santangelo, G. M. Relative Citation Ratio (RCR): A New Metric That Uses Citation Rates to Measure Influence at the Article Level. *Plos Biol* **14** (2016).
- 12 Bornmann, L., Mutz, R., Neuhaus, C. & Daniel, H.-D. Citation counts for research evaluation: standards of good practice for analyzing bibliometric data and presenting and interpreting results. *Ethics in science and environmental politics* **8**, 93-102 (2008).
- 13 Bornmann, L., Marx, W. & Barth, A. The normalization of citation counts based on classification systems. *Publications* **1**, 78-86 (2013).
- 14 Gerin, W. & Kapelewski, C. H. *Writing the NIH grant proposal: a step-by-step guide*. 2nd edn, (Sage Publications, 2011).
- 15 Pier, E. L. *et al.* Low agreement among reviewers evaluating the same NIH grant applications. *P Natl Acad Sci USA* **115**, 2952-2957 (2018).
- 16 Jacob, B. A. & Lefgren, L. The impact of research grant funding on scientific productivity. *J Public Econ* **95**, 1168-1177 (2011).
- 17 Azoulay, P., Zivin, J. S. G., Li, D. & Sampat, B. N. Public R&D Investments and Private-sector Patenting: Evidence from NIH Funding Rules. *Rev Econ Stud* **86**, 117-152 (2019).
- 18 Melin, G. & Danell, R. The top eight percent: development of approved and rejected applicants for a prestigious grant in Sweden. *Science and Public Policy* **33**, 702-712 (2006).
- 19 Bornmann, L. Scientific Peer Review. *Annu Rev Inform Sci* **45**, 199-245 (2011).
- 20 Ginther, D. K. *et al.* Race, Ethnicity, and NIH Research Awards. *Science* **333**, 1015-1019 (2011).

- 21 Ambekar, A., Ward, C., Mohammed, J., Male, S. & Skiena, S. in *Proceedings of the 15th ACM SIGKDD international conference on Knowledge Discovery and Data Mining*. 49-58 (ACM).
- 22 Iacus, S. M., King, G. & Porro, G. Causal Inference without Balance Checking: Coarsened Exact Matching. *Polit Anal* **20**, 1-24 (2012).
- 23 Iacus, S. M., King, G. & Porro, G. cem: Software for Coarsened Exact Matching. *J Stat Softw* **30**, 1-27 (2009).
- 24 Iacus, S. M., King, G. & Porro, G. Multivariate Matching Methods That Are Monotonic Imbalance Bounding. *J Am Stat Assoc* **106**, 345-361 (2011).
- 25 Weber, G. M. Identifying translational science within the triangle of biomedicine. *Journal of translational medicine* **11**, 126 (2013).
- 26 *IDENTIFYING TRANSLATION: iTrans*, <https://dpcpsi.nih.gov/sites/default/files/iTrans_one_page_04172018.pdf> (2019).
- 27 Tekles, A. & Bornmann, L. Author name disambiguation of bibliometric data: A comparison of several unsupervised approaches. *arXiv preprint arXiv:1904.12746* (2019).
- 28 Caron, E. & van Eck, N. J. in *Proceedings of the 19th international conference on science and technology indicators*. 79-86 (CWTS-Leiden University Leiden).

REVIEWERS' COMMENTS:

Reviewer #1 (Remarks to the Author):

As with their original submission, the authors were thorough in their resubmission. They more than adequately addressed my original questions and comments (the first reviewer comments in their response). I especially appreciate their efforts in exploring other funding sources for the near-win and near-miss authors. I find their approach and results almost a paper in and of itself.

The results of this paper are interesting and will generate discussion going forward. The paper is an important contribution to the literature, and I support seeing it published.

Signed: Jevin West

Reviewer #2 (Remarks to the Author):

The authors did an excellent job with the revision of the manuscript. In my opinion, not only my comments have been perfectly addressed, but also those of the other reviewers. However, one point remains which should be targeted in a further revision. In statistics, it is standard meanwhile to report and interpret not only the statistical significance of results, but also the practical significance. In other words, effect size measures should be reported and interpreted (if possible and useful). Information on effect sizes can be found in the book by J. Cohen "Statistical power analysis for the behavioral sciences" and in many statistical papers and books, for instance, in these papers: http://www.mobot.org/plantscience/ResBot/EvSy/PDF/Wilkinson_StatMeth1999.pdf; <https://psycnet.apa.org/fulltext/2017-10871-001.pdf>

Reviewer #3 (Remarks to the Author):

The authors have thoroughly addressed all my comments and successfully made the paper even stronger.

Point-By-Point Response

Referee #1

As with their original submission, the authors were thorough in their resubmission. They more than adequately addressed my original questions and comments (the first reviewer comments in their response). I especially appreciate their efforts in exploring other funding sources for the near-win and near-miss authors. I find their approach and results almost a paper in and of itself.

The results of this paper are interesting and will generate discussion going forward. The paper is an important contribution to the literature, and I support seeing it published.

Response: Thank you for appreciating our efforts. We are delighted that you support its publication.

Referee #2

The authors did an excellent job with the revision of the manuscript. In my opinion, not only my comments have been perfectly addressed, but also those of the other reviewers.

Response: Thank you very much for the positive comment about our paper. We are delighted that you find we did an excellent job addressing all reviewers' comments.

However, one point remains which should be targeted in a further revision. In statistics, it is standard meanwhile to report and interpret not only the statistical significance of results, but also the practical significance. In other words, effect size measures should be reported and interpreted (if possible and useful). Information on effect sizes can be found in the book by J. Cohen "Statistical power analysis for the behavioral sciences" and in many statistical papers and books, for instance, in these papers:

http://www.mobot.org/plantscience/ResBot/EvSy/PDF/Wilkinson_StatMeth1999.pdf;

<https://psycnet.apa.org/fulltext/2017-10871-001.pdf>

Response: Thank you very much for this helpful comment. Follow your suggestion, in the revised manuscript, we have also reported the standardized effect sizes together with the unstandardized effect sizes whenever necessary.

Referee #3:

The authors have thoroughly addressed all my comments and successfully made the paper even stronger.

Response: Thank you very much. We are delighted that you find the paper even stronger.